# Substantial terrestrial carbon emissions from global expansion of impervious surface area

Linghua Qiu [1,2,7], Junhao He[3,7], Chao Yue [1,3,4] ✉, Philippe Ciais[5] & Chunmiao Zheng [2,6]

Global impervious surface area (ISA) has more than doubled over the last three decades, but the associated carbon emissions resulting from the depletion of pre-existing land carbon stores remain unknown. Here, we report that the carbon losses from biomass and top soil (0–30 cm) due to global ISA expansion reached 46–75 Tg C per year over 1993–2018, accounting for 3.7–6.0% of the concurrent human land-use change emissions. For the Annex I countries of UNFCCC, our estimated emissions are comparable to the carbon emissions arising from settlement expansion as reported by the national greenhouse gas inventories, providing independent validation of this kind. The contrast between growing emissions in non-Annex I countries and declining ones in Annex I countries over the study period can be explained by an observed emerging pattern of emissions evolution dependent on the economic development stage. Our study has implications for international carbon accounting and climate mitigation as it reveals previously ignored but substantial contributions of ISA expansion to anthropogenic carbon emissions through land-use effects.

The global urban land area has more than doubled over the past three decades[1] and will continue to expand, by at least a half, or as much as another doubling, until the end of this century[2]. The impervious surface area (ISA), consisting of pavements, roads, built-up areas, squares and parking lots, etc., with ground coverage of tar, concrete, asphalt or mixtures of such materials, is a core feature of urban land enabling its identification through remote sensing[1,3–6]. Because of the distinctive land surface characteristics of ISA compared to vegetated land, its global expansion has profound environmental effects, including the creation of heat islands[7,8], loss of biodiversity[4,9], and perturbations of the carbon[4], nitrogen[10] and hydrological cycles[11].

Replacing carbon-rich ecosystems with plant-free ISA leads to a loss of carbon in both biomass and soil[4], but as yet, such carbon losses have not been addressed in assessments of the global carbon budget made by the Global Carbon Project (GCP)[12]. In the GCP, carbon emissions from land use and land use change have been estimated using two main approaches: bookkeeping models and dynamic global vegetation models (DGVMs). Transitions from vegetated land to ISA were not included in any of the three bookkeeping models used in the GCP carbon budget assessments[12], which were further adopted in the 6th Assessment Report (AR6) of the Intergovernmental Panel on Climate Change (IPCC). In addition, the representation of ISA (or urban land) is inconsistent among DGVMs, with it being either treated as

[1]Shenzhen Research Institute, Northwest A&F University, 518000 Shenzhen, China. [2]School of Environmental Science and Engineering, Southern University of Science and Technology, 518055 Shenzhen, China. [3]State Key Laboratory of Soil Erosion and Dryland Farming on the Loess Plateau, Institute of Soil and Water Conservation, Northwest A&F University, 712100 Yangling, Shaanxi, China. [4]College of Natural Resources and Environment, Northwest A & F University, Yangling, Shaanxi, China. [5]Laboratoire des Sciences du Climat et de l'Environnement, LSCE/IPSL, CEA-CNRS-UVSQ, Université Paris-Saclay, 91191 Gif-sur-Yvette, France. [6]Eastern Institute for Advanced Study, Eastern Institute of Technology, 315200 Ningbo, China. [7]These authors contributed equally: Linghua Qiu, Junhao He. ✉e-mail: chaoyuejoy@gmail.com

pastureland or cropland, or bare land with no carbon fluxes (Supplementary Table 1). This neglect of carbon emissions from ISA expansion in both bookkeeping models and DGVMs demonstrates that these emissions are a missing component of the current global carbon budget assessment.

Although a few studies have quantified the loss of terrestrial net ecosystem productivity (NEP) or net primary productivity (NPP) caused by global ISA[13] or urban[14] expansion, the lost NEP or NPP represents an amount of unrealized carbon uptake, which adds no physically tangible $CO_2$ to the atmosphere. Terrestrial carbon losses from historical ISA expansion (hereafter referred to as ISA-driven carbon emissions), a potentially important component of anthropogenic carbon emissions contributing to the atmospheric $CO_2$ growth, remain largely ignored by the global carbon-cycle science community.

On the other hand, according to IPCC guidelines, the 'settlements' land use type−all human-developed land including residential, transportation, commercial, and production infrastructure−is one of the six key types for which nations should report anthropogenic carbon losses and/or gains in the land use sector[15]. Carbon emissions from human settlement expansion have been reported in national greenhouse gas inventories (NGHGIs) by Annex I (AI) countries of the United Nations Framework Convention on Climate Change (UNFCCC) for the last 30 years. This 'settlement' land includes, by definition, both ISA and the associated vegetated land (e.g., garden plants)[16]. However, data reported by AI countries show that the carbon effects due to settlement expansion are dominated by carbon losses from ISA expansion, rather than by carbon gains from vegetated areas within settlements (Supplementary Fig. 1). This justifies using independent satellite-based estimates of ISA-driven carbon emissions to validate the reported carbon losses due to settlement expansion by NGHGIs. However, to the best of our knowledge, no such validation has been reported.

This study aims to provide an estimate of the terrestrial carbon losses from direct land use effects of global ISA expansion over 1993−2018 (see "Methods"). To reduce the uncertainty that would result from using a single dataset, we used four state-of-the-art global remote-sensing products of impervious surface area: three products with a 30-m resolution (GAUD[1], GAIA[6] and GISA[17]) along with the 300-m resolution ESA CCI product[18]. The ESA CCI product was included, despite its medium resolution, for two reasons: (1) the amount and spatial patterns of ISA expansion provided by ESA CCI are similar to the other three high-resolution products (Supplementary Fig. 2); (2) the ESA CCI product has commonly been used to identify source land cover of ISA (or urban) expansion in previous studies[1,2].

Our analysis focuses on the carbon effects of ISA expansion by assuming 100% ISA coverage for the mapped ISA pixels by all four products. Any potential sub-pixel green areas (e.g., street trees, residential gardens) are thus ignored. Such an approach is consistent with the landscape homogeneity assumption that underlies the classification of land cover and its subsequent spatial mapping (Supplementary Information 1). The same approach has been adopted in previous studies investigating global and regional ISA expansion and the associated effects on the land carbon cycle[4,13,14]. We argue that this is an imperfect but pragmatic approach that will unlikely lead to an overall overestimation of global ISA expansion, especially considering that ISA alongside global minor roads might not have been included in the ISA products (Supplementary Information 1).

The individual ISA products were used to derive annual global ISA expansions, with ISA source land covers being obtained from the ESA CCI land cover product. For soil organic carbon (SOC) loss following ISA establishment, we used two alternative loss ratios: the IPCC default value (lower boundary) and the value derived from a literature synthesis in this study (upper boundary). Carbon losses due to conversion from vegetated land to ISA were then quantified by overlaying the spatial maps of ISA expansion, source land covers, and the carbon

stock maps of above- and below-ground live biomass, surface litter, dead wood and SOC[19–22]. Biomass losses were limited to source land covers of forest, shrubland, wetland, cropland, and grassland, whereas SOC losses were assumed for all source land covers. The quantified losses represent committed emissions because the carbon was assumed to be lost immediately upon conversion to ISA: a common practice when evaluating land-use carbon emissions based on remote sensing[4,21]. The derived ISA-driven carbon emissions were compared with those due to settlement expansion as reported by NGHGIs for AI countries. Finally, we employed a structural decomposition framework and attributed the emission dynamics over 1993−2018 to different underlying factors of socioeconomic and urban dynamics. This attribution analysis was carried out for the entire globe and then separately for the AI and Non-Annex I (NAI) country groups. Full details are provided in the "Methods" section.

## Results
### Global ISA expansion and the associated carbon losses over 1993–2018

Averaging the four ISA products reveals that the global ISA has been expanding at a rate of $15,913 \pm 3331$ km² yr⁻¹ over 1993−2018 (Fig. 1a). The cumulative ISA expansion reached 0.41 Mkm² (Mkm² = 10⁶ km²) during the study period (Fig. 1a), which represents an increase of 112% of global ISA, from 0.37 Mkm² in 1992 to 0.78 Mkm² in 2018. About 65% of this expansion took place at the cost of cropland, while 12%, 11%, 4%, and 1% replaced forest, grassland, shrubland, and wetland, respectively (Fig. 1b). Considering the individual ISA products, the expansion rate ranges from 13,948 km² yr⁻¹ in GAUD to 16,956 km² yr⁻¹ in GAIA, being 4.4−5.3 times larger than the urban area growth rate of 3173 km² yr⁻¹ previously reported using the 500-m resolution MODIS land cover products[23]. Three of the ISA products show accelerating ISA expansion over time, while GAUD shows a deceleration of −220.7 km² yr⁻² ($p = 0.10$). The average acceleration rate over 1993−2018 is 305.5 km² yr⁻² ($p < 0.01$).

Our global-scale meta-analysis using the 22 available paired measurements of SOC under urban green areas and ISA showed an average loss ratio of $59.5 \pm 16.6\%$ in SOC following ISA establishment (see "Methods"). This value is notably higher than the default loss ratio of 20% recommended in the Tier 1 approach of the IPCC guidelines for NGHGIs but the latter has an unknown level of support from field observations. Nonetheless, given the widespread application of the default value of IPCC, our derived SOC loss ratio of 59.5% and the IPCC default value of 20% are used as the upper and lower boundaries, respectively, of the SOC loss ratio for quantifying SOC losses from the direct land use effects of ISA expansion (see "Methods").

By averaging the four ISA products, the terrestrial carbon losses caused by ISA expansion with the upper boundary of SOC loss reached $74.9 \pm 13.7$ Tg C yr⁻¹ for 1993−2018, with the values of the individual ISA products ranging from 65.2 Tg C yr⁻¹ for GAUD to 81.4 Tg C yr⁻¹ for ESA CCI (Fig. 1c). The average carbon emissions with the lower boundary of SOC loss were $45.8 \pm 8.2$ Tg C yr⁻¹, with the values of individual ISA products ranging from 39.9 Tg C yr⁻¹ for GAUD to 49.8 Tg C yr⁻¹ for ESA CCI (Fig. 1c). Loss of soil organic carbon made up 32−59% of the total emissions, with the remaining (41−68%) coming from the biomass carbon stocks of above- and below-ground live biomass, surface litter, and dead wood (Fig. 1d). The ISA-driven biomass emissions were $31.0 \pm 5.5$ Tg C yr⁻¹ over 1993−2018 by averaging the four ISA products. In terms of source land cover distribution, 49−55% of the emissions resulted from the conversion of cropland, while conversion of forest, grassland, shrubland, and wetland accounted for 25−31%, 11−12%, 4−5%, and 1%, respectively (Fig. 1e, f).

Concurrent with the overall acceleration in global ISA expansion over time, the annual carbon loss calculated by averaging the emissions derived from all ISA products showed a small trend of 0.36−0.78 Tg C yr⁻² ($p = 0.10$/$p < 0.05$) over 1993−2018 (Fig. 1c). Such a global

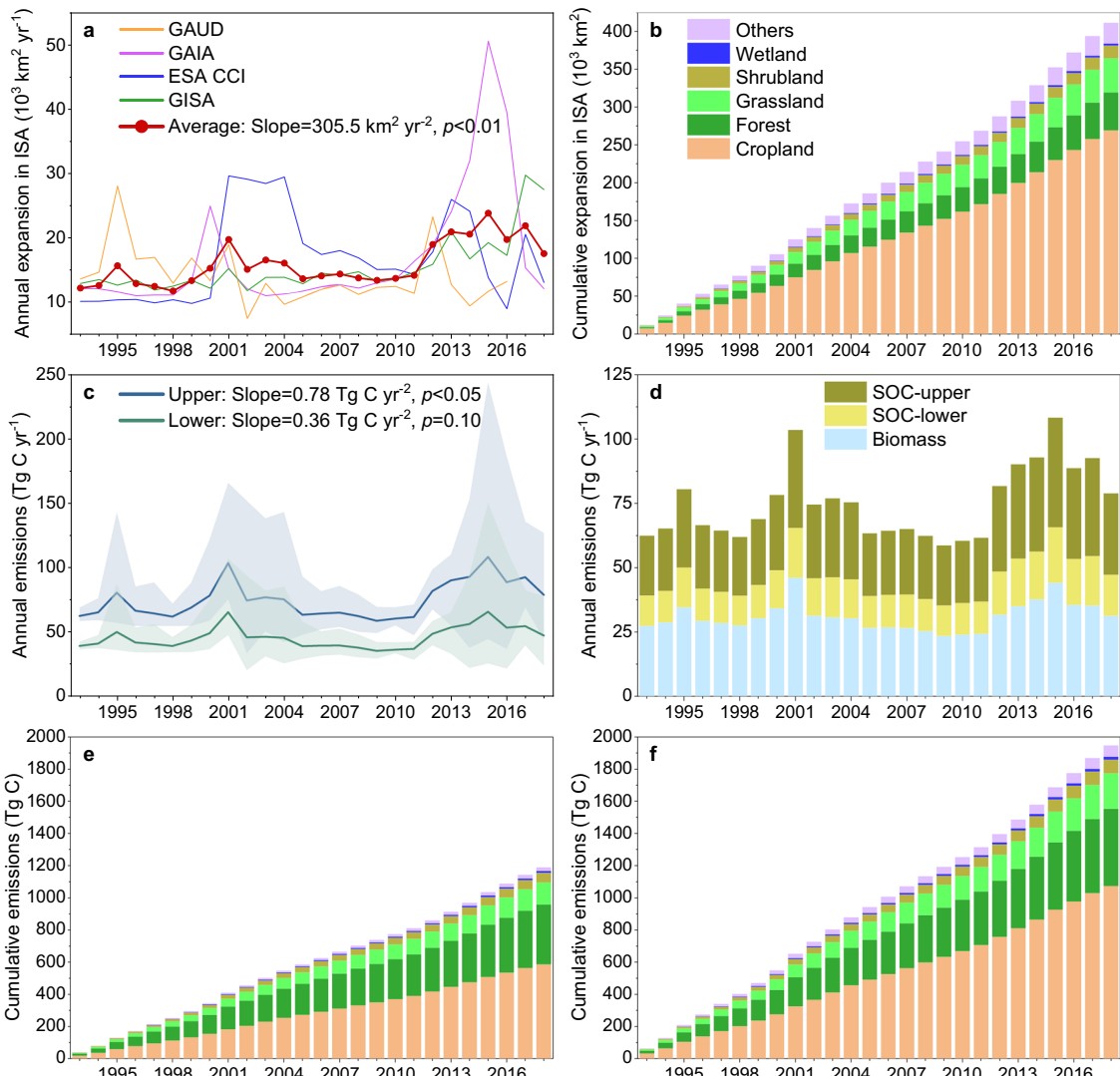

**Fig. 1 | Global impervious surface area (ISA) expansion and the associated carbon emissions over 1993–2018. a** Global annual increases in impervious surface area. **b** Cumulative increases in ISA from different land cover sources. **c** Annual carbon losses from direct land-use effects of ISA expansion with upper and lower boundaries. **d** Annual carbon losses from biomass (including live biomass, surface litter and dead wood) and soil organic carbon, respectively. Please note that stacked bars of soil organic carbon (SOC) emission with the upper and lower boundary are both based on the same biomass bars. **e** and **f** represent cumulative carbon losses from different source land covers with the lower and upper boundary, respectively. The red thick line with dots in panel (**a**) shows the average of the results from the four ISA products (i.e., GAUD, GAIA, ESA CCI and GISA), which is also used in panels (**b–f**). Shaded regions around the lines in panel (**c**) show the range of the estimates from the four ISA products. Panels (**b**), (**e**) and (**f**) share the same legend.

trend is a result of a significantly positive trend for the NAI countries (0.53–1.01 Tg C yr$^{-2}$, $p < 0.01$) being partly offset by a negative, albeit not statistically significant, trend for the AI countries (−0.23 to −0.17 Tg C yr$^{-2}$, $p > 0.05$) (Supplementary Fig. 3). These contrasting trends for the two country groups are linked to their respective ISA expansion trends, with the NAI countries having a strong increasing trend over time (320.9 km$^2$ yr$^{-2}$, $p < 0.01$), which, again, is partly offset by a decreasing trend for the AI countries (−15.4 km$^2$ yr$^{-2}$, $p > 0.05$).

The 15 countries or regions (the 27 countries of the European Union were treated as the single unit: EU27) with the largest emissions accounted for 81.1–82.0% of global total emissions over 1993–2018 (Fig. 2a, b), with their growths in urban population (ΔPu) and ISA (ΔISA) accounting for 63.3% and 79.3% of the global totals, respectively. Distinct patterns were found among the large emitters. ISA expansion and the resultant carbon emissions in China, Brazil, India and Indonesia were mainly driven by urban population growth, with 42, 51, 104 and 128 new urban residents per hectare of new ISA (ΔPu/ΔISA), respectively. In contrast, emissions in the USA, EU27, Japan, Canada, Australia,

and the UK were driven more by the growth in ISA, with relatively small urban population growth of 11, 8, 26, 16, 15 and 28 new urban residents per hectare of new ISA, respectively. Collectively, the AI countries accounted for 10.1% of the global urban population growth but 39.2% of ISA expansion and 51.1–54.5% of carbon emissions (Fig. 2a, b), highlighting their disproportionately large emissions compared to their urban population growth.

The spatial distribution of carbon emissions largely follows that of ISA expansion (Fig. 2c–e), showing pronounced patterns driven by the development of urban clusters across different nations. For example, in China, emission hotspots were found in the central plain in northern China, including the urban clusters of Beijing-Tianjin-Hebei, the Yangtze River Delta urban cluster, the Pearl River Delta urban cluster, and two inland urban clusters in western China centered on Xi'an and Chengdu-Chongqing. In India, the urban belt comprising Delhi, Lucknow, Patna, and Kolkata shows prominent emissions. In Russia and central and northern Europe, ISA-driven carbon emissions were more pronounced than ISA expansion because of their higher emissions per

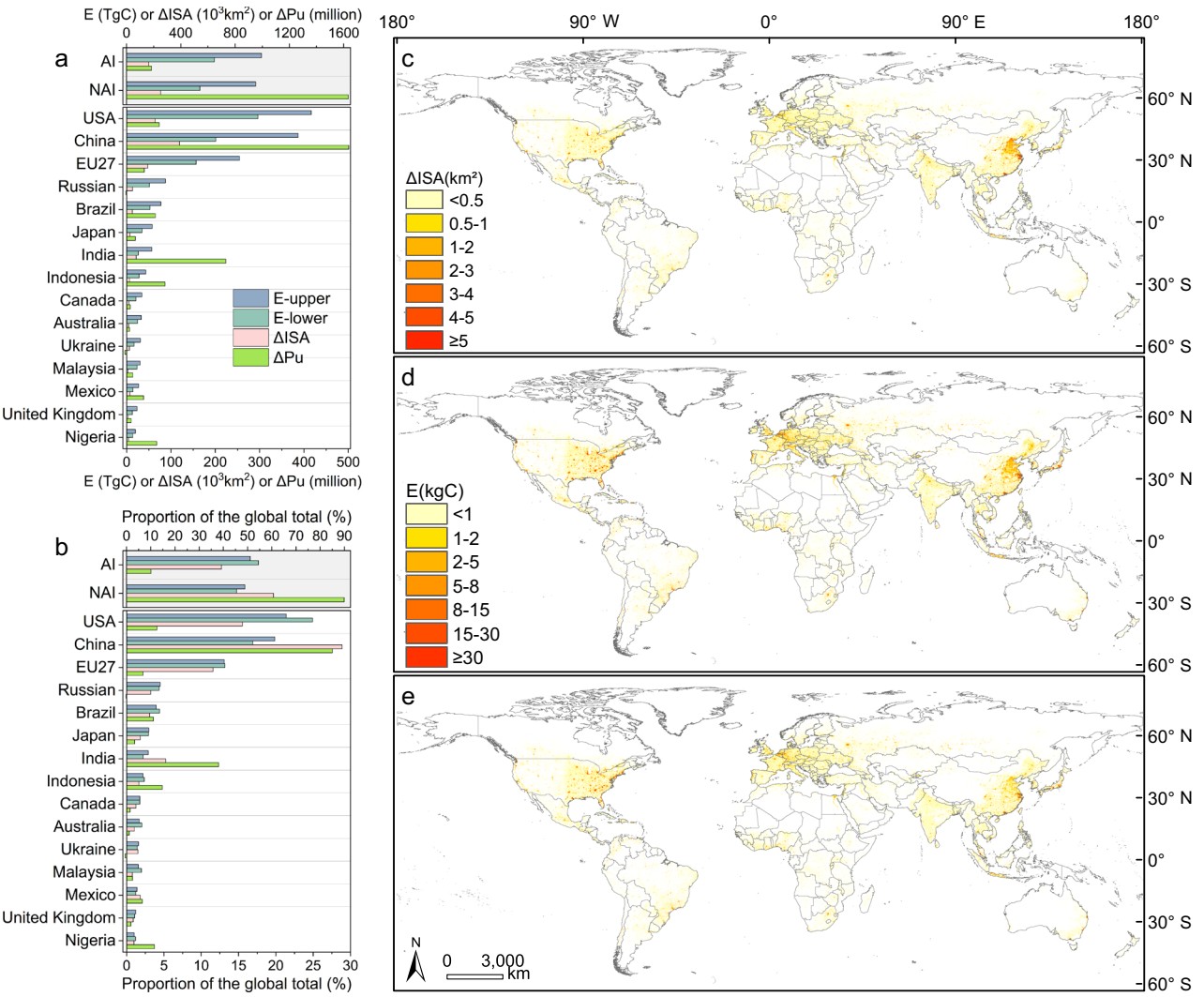

**Fig. 2 | Impervious surface area (ISA) expansion and the associated carbon emissions during 1993–2018 for leading countries and country groups, and their spatial distributions. a** Cumulative emissions (E), the increases in urban population (ΔPu) and in ISA (ΔISA) are shown for Annex I (AI) and Non-Annex I (NAI) country groups, as well as for the 15 countries with the largest emissions. EU27 is the 27 countries of the European Union, which were treated as the single unit. **b** The percentage contributions of each country or country group to the global total value. **c** Spatial distribution of the cumulative ISA expansion, shown at a nominal 5-km resolution. **d** Spatial distribution of the associated carbon emissions with the upper boundary, shown at a nominal 5-km resolution. **e** Same as panel (**d**) but with the lower boundary. Panels (**d**) and (**e**) share the same legend.

ΔISA, which were mainly driven by greater emissions from SOC (Supplementary Fig. 4).

## Comparison with carbon emissions due to human settlement expansion for Annex I countries

The derived ISA-driven carbon emissions, including both biomass and SOC, were compared with those due to land conversion to settlements as reported by NGHGIs for Annex I countries. Annual carbon losses for biomass (including living biomass, surface litter, and dead wood) estimated in this study (18.2 ± 3.9 Tg C yr⁻¹) were close to those reported by NGHGIs (22.9 ± 0.9 Tg C yr⁻¹) (Fig. 3a). Given that 'human settlements' in NGHGIs can contain vegetated land and that the definitions are at the discretion of each nation, not completely to our surprise, during 1993–2018, the land area converted to settlements reported by AI countries (82,596 ± 13,864 km² yr⁻¹) was an order of magnitude larger than the ISA expansion obtained in this study (6245 ± 1282 km² yr⁻¹) (Supplementary Fig. 5a). The comparable estimates of biomass carbon losses between this study and the NGHGIs thus imply that substantial areas considered as 'settlements' by NGHGIs contributed little to biomass carbon gain or losses. Indeed,

gross carbon gains in live biomass (excluding surface litter and dead wood) due to settlement expansion reported by NGHGIs were as small as 2.6 ± 0.6 Tg C yr⁻¹, accounting for only one-seventh of the reported gross biomass carbon losses (19.2 ± 1.0 Tg C yr⁻¹) (Supplementary Fig. 1). This implies that, in terms of carbon stocks, the vast potentially vegetated areas excluded from our analysis, but considered as settlements by NGHGIs, differ only slightly from their source land covers.

The annual SOC loss reported by NGHGIs (14.8 ± 2.2 Tg C yr⁻¹) falls between our lower (6.8 ± 1.5 Tg C yr⁻¹) and upper (20.1 ± 4.3 Tg C yr⁻¹) boundaries of estimate (Fig. 3b). Annex I countries applied either the Tier 1 approach (i.e., a default SOC loss ratio of 20%) or a higher-Tier approach based on country-specific data (e.g., the USA used a SOC loss ratio of 30%), and thus the default loss ratio of 20% is likely the minimum applied in NGHGIs. This justifies our decision to use it in our lower-boundary estimation and explains the fact that the NGHGI results fall between our lower and upper boundaries. As an attempt to evaluate SOC losses from global ISA expansion by means of a meta-analysis, despite a limited number of publications, our results highlight the need for more observations on ISA-driven SOC loss and for the continual update of the SOC loss ratio in the IPCC guidelines.

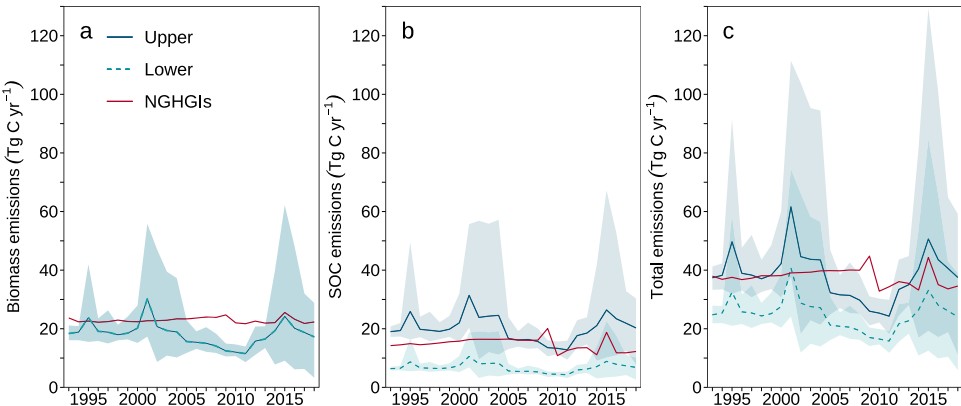

**Fig. 3 | Carbon losses due to impervious surface area (ISA) expansion estimated by this study and from land conversion to settlements as reported in national greenhouse gas inventories (NGHGIs) for Annex I countries. a** Biomass carbon emissions (including living biomass, surface litter and dead wood). **b** soil organic carbon (SOC) emissions. **c** Total carbon emissions including biomass and SOC. Solid blue line and dashed green line represent the mean carbon losses estimated from four ISA products by adopting the upper and lower SOC loss ratios in this study, respectively. Red solid line represents the carbon losses reported in NGHGIs for Annex I countries. Shaded regions around the solid blue and dashed green lines show the range of the estimates from the four ISA products.

Overall, the annual carbon losses reported by NGHGIs (37.8 ± 2.9 Tg C yr$^{-1}$) between 1993 and 2018 were higher than our estimate of the lower boundary (24.9 ± 5.3 Tg C yr$^{-1}$) but close to the upper boundary (38.3 ± 8.2 Tg C yr$^{-1}$) (Fig. 3c), despite the large area of land reported having converted to settlements. Interestingly, the USA had the single largest increase in settlement area of any of the AI countries, accounting for 90.4% of the total increase of this group, but contributed only about a half of the group's carbon emissions (Supplementary Fig. 5b). Excluding the USA led to both our quantified ISA expansion and the associated emissions being closer to those given by the NGHGIs, with the area of ΔISA being 3776 ± 1071 km$^2$ yr$^{-1}$ and settlement expansion being 7895 ± 1150 km$^2$ yr$^{-1}$ (Supplementary Fig. 5a). The carbon emissions given by our lower and upper boundaries were 13.5 ± 3.7 Tg C yr$^{-1}$ to 22.3 ± 6.0 Tg C yr$^{-1}$, respectively, while those given by the NGHGIs were 17.2 ± 3.0 Tg C yr$^{-1}$ (Supplementary Fig. 5b).

## Socioeconomic drivers for ISA-driven carbon losses

We developed an "ISA-driven Emissions Identity" framework, in analog to the widely used Kaya Identify framework in energy $CO_2$ emissions attribution[24,25], to attribute the relative change rate in ISA-driven carbon emissions ($E$) to the different driving factors of total **p**opulation ($P$), **u**rbanization rate ($u$, the ratio of urban population ($Pu$) to $P$), **r**esidential ISA intensity ($r$, the ratio of the existing $ISA$ to $P_u$), ISA expansion **s**peed-up factor ($s$, the ratio of annually expanding ISA ($\Delta ISA$) to $ISA$), and the carbon **e**mission intensity ($e$, the ratio of $E$ to $\Delta ISA$) (for details see "Methods"). The emission dynamics, according to both our lower- and upper-boundary estimates, were individually attributed to these driving factors by applying this decomposition framework (Supplementary Fig. 6).

Given that similar patterns in the various driving factors were obtained using the lower- and upper-boundary emissions (Supplementary Fig. 7), the average results of the two attributions are shown in Fig. 4. The temporal trend of 0.78 (0.36) Tg C yr$^{-2}$ in the global ISA-driven carbon emissions ($E$) with the upper (lower) boundary translates into a relative growth rate of 1.05% (0.79%) per year over 1993–2018 and an average growth rate of 0.92% (Fig. 4). The attribution results show that such a small growth rate is an outcome of two groups of driving factors with opposing effects: the relative growth in the global population (1.25% yr$^{-1}$ in $P$), urbanization rate (0.93% yr$^{-1}$ in $u$), and residential ISA intensity (0.64% yr$^{-1}$ in $r$) added to a gross growth rate of 2.82% in $E$, which was offset by a negative growth rate of −0.99% per year in $s$ (i.e., global ISA expansion has decelerated over time) and −1.00% yr$^{-1}$ in $e$ (i.e., emission intensity) (Fig. 4, Supplementary Figs. 6 and 7).

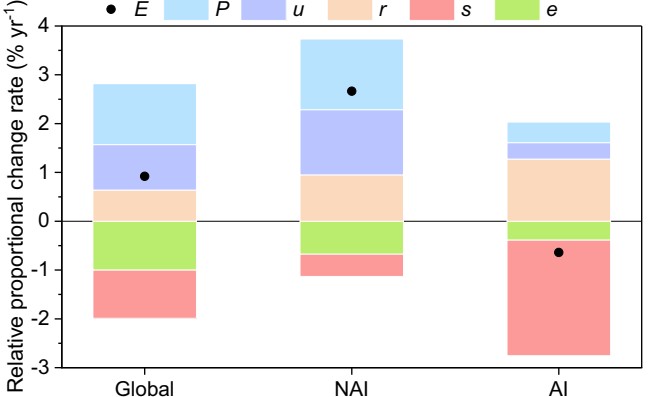

**Fig. 4 | Drivers of the dynamics in carbon emissions due to impervious surface area (ISA) expansion over 1993–2018.** Relative proportional change rates in ISA-driven carbon emissions ($E$) and the underlying drivers (total population $P$, urbanization rate $u$, residential ISA intensity $r$, ISA expansion speed-up factor $s$, and the carbon emission intensity $e$) are shown for the global total, the non-Annex I countries (NAI), and the Annex I countries (AI) of the United Nations Framework Convention on Climate Change (UNFCCC). The average results of the upper- and lower-boundary estimates of carbon emissions are shown. The individual attribution results for the upper- and lower-boundary carbon emissions are shown in Supplementary Fig. 7. The annual time series of all variables for 1993–2018 are shown in Supplementary Fig. 6.

These results suggest that global population growth, urban migration and the increasing per-capita ISA demand have been driving ISA expansion over the past three decades, but the growth rate of ISA has been slowing with time. At the global scale, the carbon emission intensity shows a decreasing trend, being −0.41 Mg C ha$^{-1}$ yr$^{-1}$ ($p < 0.01$) for the upper boundary and −0.33 Mg C ha$^{-1}$ yr$^{-1}$ ($p < 0.01$) for the lower boundary, which consists of both decreasing biomass and SOC emission intensities (Supplementary Fig. 8). These decreasing emission intensities of biomass and SOC are partly caused by the declining ratio of woody lands to all vegetated land sources of ISA expansion because woody vegetation (forest and shrubland) tends to have a greater biomass and SOC density than other herbaceous vegetations (Supplementary Fig. 8). Further analysis reveals that there has been a shift in the latitudinal distribution of ISA expansion during the study period (Supplementary Fig. 9). During the early years, the distribution is dominated by the northern mid-to-high latitudes, where both biomass and SOC emissions densities are high. However, in the later years of the

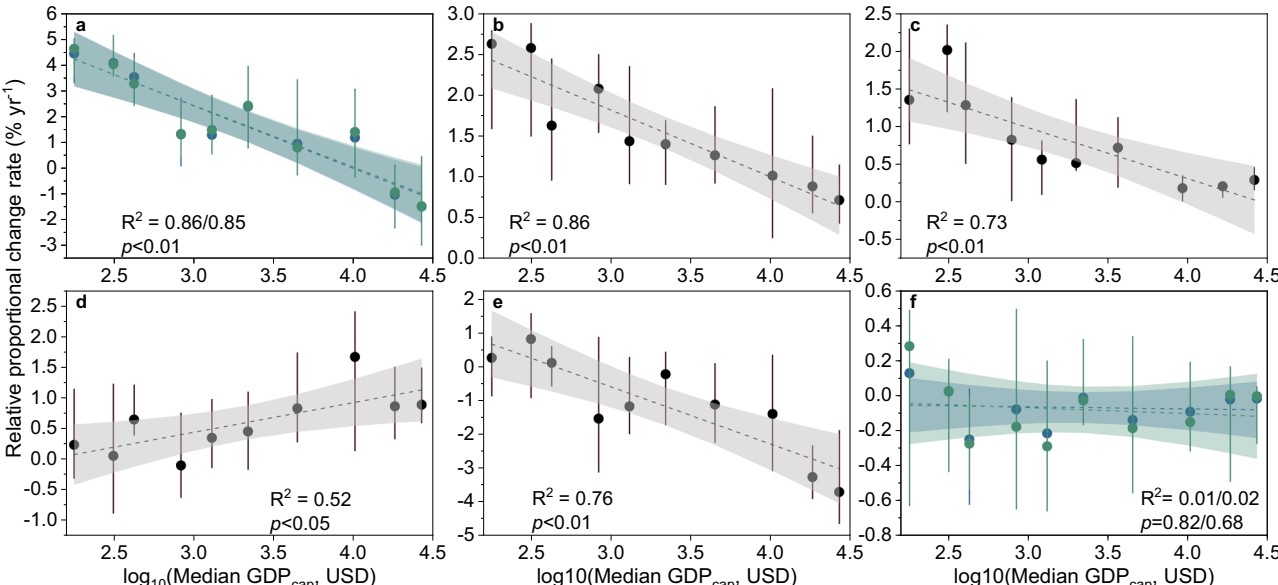

**Fig. 5 | Relationships between relative proportional change rates in impervious surface area (ISA) driven carbon emissions and the underlying drivers over 1993–2018 and per capita gross domestic product (GDP). a** The relative change rate in carbon emissions (*E*). **b**–**f** The relative change rates in total population *P*, urbanization rate *u*, residential ISA intensity *r*, ISA expansion speed-up factor *s*, and the carbon emission intensity *e*, respectively. Solid dots represent the median values of proportional change rates in the carbon emissions and their driving factors over 1993–2018 for the ten different country groups of increasing per capita GDP (GDP$_{cap}$, see "Methods"), with vertical lines showing 25–75% percentiles. The horizontal axis shows the logarithmically transformed median values of GDP$_{cap}$ in 1993 for each country group. The dashed lines represent fitted linear regressions, with the shading indicating the 95% confidence intervals. For panels (**a**) and (**f**), light blue indicates the results using the upper-boundary emissions, while light green indicates those with the lower-boundary emissions.

study period, the subtropical northern hemisphere becomes dominant. Since both emissions densities are comparatively low in the subtropical northern hemisphere, this change in the spatial distribution of ISA expansion has additionally contributed to the declining emissions densities.

The contrasting temporal trends in emissions between AI (declining) and NAI (growing) countries can also be explained by their respective changes in the underlying driving factors. Averaging the upper- and lower-boundary results reveals that carbon emissions in NAI countries had a relative growth rate of 2.66% yr$^{-1}$, whereas those in AI countries showed a decreasing rate of −0.64% yr$^{-1}$ (Fig. 4). The relative growth in total population (*P*) was higher in NAI countries (1.45% yr$^{-1}$) than AI ones (0.42% yr$^{-1}$), likely implying a higher demand for ISA. This difference is further enhanced by an increase in the urbanization rate (*u*) in NAI countries (1.34% yr$^{-1}$), which is an order of magnitude larger than that in AI countries (0.34% yr$^{-1}$). In contrast, the growth in residential ISA intensity (*r*) was higher in AI countries (1.27% yr$^{-1}$) than in NAI ones (0.95% yr$^{-1}$). Although ISA expansion has been decelerating for both country groups over 1993–2018, the deceleration was faster in AI countries (−2.37% yr$^{-1}$) than in NAI ones (−0.45% yr$^{-1}$).

Given that economic development is generally more advanced in AI countries than in NAI ones, the contrasts in emissions dynamics between the two country groups likely suggest a hypothesized pattern in emissions evolution which is dependent on nations' economic development state. That is, as the economy develops with time, the growth in total population and urban migration will slow but the growth in residential ISA intensity will accelerate. These changes, combined with a decelerating ISA expansion rate, lead to growing emissions in the early stages of economic development but declining ones in the late stage. To confirm this hypothesis, we used the per capita gross primary production (GDP$_{cap}$) in 1993 as an indicator for economic development and investigated the relationships between the relative change rate of *E*, and its driving factors, over 1993–2018

and GDP$_{cap}$ at country scales (see "Methods"; using the 2018 GDP$_{cap}$ yielded similar results, see Supplementary Fig. 10).

The results confirm the pattern hypothesized above by revealing consistent dependencies of emissions dynamics and their drivers on GDP$_{cap}$. The relative change rates of carbon emissions across different country groups significantly decreased with log$_{10}$(GDP$_{cap}$) (Fig. 5a), with high growth rates in *E* being found in countries with low GDP$_{cap}$ and, vice versa, lower growth rates in countries with higher GDP$_{cap}$, until *E* starts to decline in countries with a GDP$_{cap}$ exceeding ca. 10,000 US$ (at 2022 price). The relative change rates in total population (*P*), urbanization rate (*u*), and ISA expansion speed-up factor (*s*) all show significantly decreasing relationships with log$_{10}$(GDP$_{cap}$) (Fig. 5b, c and e), but the relative change rate in residential ISA intensity (*r*) increases with log$_{10}$(GDP$_{cap}$) (Fig. 5d). The relative change rate in emission intensity (*e*) shows no significant relationship with log$_{10}$(GDP$_{cap}$) (Fig. 5f), which is plausible because emission intensity largely depends on the carbon stock density of source land cover for ISA, which further depends on the natural conditions of different countries rather than on their economic development state.

## Discussion

We report a substantial contribution of global ISA expansion to anthropogenic carbon emissions that has been overlooked in previous global carbon budget assessments. The estimated ISA-driven emissions of 1.19–1.95 Pg C over 1993–2018 from biomass and top soil (0–30 cm), obtained by averaging the results of the four ISA products, account for 3.7–6.0% of the global land-use change emissions[12,26]. In fact, the percentage contribution of ISA-driven emissions to global land-use change emissions could be an underestimate because, in the latter, the loss in SOC up to a depth of 1 m was used. Assuming the same SOC loss ratio for the 0–1 m soil layer, rather than for top soil only, increased the percentage further to 6.0–12.9%. Our results thus underline the profound effect of global ISA expansion on global carbon cycles and, further, on global climate change.

This study focuses on terrestrial carbon losses from ISA expansion by assuming 100% ISA coverage for the mapped ISA pixels of all four products (for a discussion of the validity of this assumption, refer to Supplementary Information 1). The non-ISA but vegetated pixels within 'urban environment', broadly termed 'urban vegetation' or 'urban green space', possibly including urban forest, lawn, urban wetland and cropland, are hence excluded from our scope. This intentional exclusion of urban vegetation is based on two considerations: (1) Carbon balances of large tracts of urban vegetation identified as non-ISA pixels in land cover products have been largely accounted for in existing bookkeeping models or DGVMs in the current global carbon budget accounting, whereas the objective of this study is to account for the neglected carbon effects of ISA expansion; (2) To-date, no commonly shared definition of "urban" or "urban vegetation" exists within the academic community, leading to great uncertainty in estimates of the area of urban vegetation. The proportion of urban area to the world's total land area ranges from 0.2 to 3%[27] according to different definitions, thereby preventing any sensible estimation of carbon stock changes over urban vegetation using a bottom-up approach.

A few studies have reported that direct human management of vegetation, such as irrigation, pest control, and fertilizer application, can enhance urban vegetation greenness, potentially leading to enhanced photosynthetic carbon uptake[28,29]. Indirect effects of urban development, such as nutrient deposition and the heat-island effect, also might enhance urban vegetation growth[30]. Enhanced urban vegetation productivity further increases plant carbon inputs into the soil, resulting in SOC accumulation over urban vegetation[31]. Hence, if this study is extended to a broader scope of 'urban area' covering both ISA and urban vegetation, then part of the ISA-driven terrestrial losses could be offset by carbon sinks over urban vegetation. Unfortunately, to our knowledge, robust estimates of any substantial carbon sink in urban vegetation remain absent on a large spatial scale (partly due to the uncertain area of urban vegetation), making estimating the size of the offset effect highly challenging. Nonetheless, the national inventories of AI countries show a small, almost negligible, live biomass carbon sink due to land conversion to settlements, which includes both ISA and urban vegetation. This result likely indicates a small contribution of carbon sequestration over urban vegetation to offset ISA-driven carbon losses.

The need to incorporate satellite-based information on land use change into NGHGIs has been recently advocated and highlighted in a global effort known as the REgional Carbon Cycle Assessment and Processes (RECCAP2), part of the Global Carbon Project[32]. In addition, satellite-based evaluations of land use change emissions contribute to verifying the greenhouse gas emissions reported by nations and to enhancing their transparency. Such verification and transparency are key requirements for implementing the Paris Agreement for global temperature control[33,34]. The carbon emissions estimated in this study for the Annex I countries of UNFCCC are largely comparable with those due to settlement expansion reported in the NGHGIs and thus provide an independent evaluation of the terrestrial carbon effects associated with human settlements given in national inventories. In addition, our bottom-up estimates of ISA-driven carbon emissions, made by applying a consistent framework, can also contribute to the upcoming Global Stocktake (starting in 2023 and due to be made every 5 years thereafter) under the Paris Agreement. In particular, our estimates may potentially contribute to the NGHGIs of NAI countries which are obliged to provide detailed greenhouse inventories under the Paris Agreement but are often limited by their capacities[34].

Our analysis also reveals a clear pattern of ISA-driven emissions dynamics along with economic development, establishing a predictive framework for future emissions driven by economic and social developments. A recent study projected that the global urban area would increase by 0.35−0.69 Mkm² by the end of this century under different levels of urbanization driven by global population and economic development under different socioeconomic shared pathways[2]. Using business-as-usual techniques to expand impervious surface areas for this anticipated urban growth will cause additional carbon losses[35]. Global urbanization has long been depicted as the rural-urban migration of the population, with ISA expansion being undervalued, particularly in the context of urban policymaking[36]. Discussions on climate mitigation potential in urban areas also often focus on their energy systems while largely neglecting the land sector[37–39]. Our study highlights the fact that reducing terrestrial carbon losses through sustainable urban planning remains a great challenge and deserves attention from the public, urban planners, policymakers, and urban stakeholders.

## Methods

### General overview of the methodology

This study aims to quantify terrestrial carbon losses from global impervious surface area (ISA) expansion (referred to as ISA-driven emissions) over 1993–2018. Four state-of-the-art global remote-sensing products of impervious surface or urban area were used. Our analysis focuses on the dynamics of impervious surface area and assumes 100% coverage of ISA for the identified ISA (or urban) pixels in the ISA (or urban) products (hereafter referred to as ISA products) at their original spatial resolutions by ignoring any potential sub-pixel, urban green areas (e.g., street trees, residential gardens, etc., see Supplementary Information 1 for justification of our decision). The four ISA products were used to derive individual ISA expansion rates, whose average value was then obtained and examined. ESA CCI land cover products were used to obtain source land cover information for global ISA expansion on an annual time scale.

Carbon density maps of various carbon pools, consisting of living biomass (both above- and below-ground), surface litter, dead wood, and soil organic carbon (SOC), were then overlaid with the maps of ISA expansion derived from the different ISA products to quantify terrestrial carbon losses. Immediate carbon losses on conversion of vegetated lands to ISA were assumed, and hence the quantified carbon losses were committed emissions. The estimated ISA-driven emissions were further compared with carbon losses due to land conversion to settlements as reported in national greenhouse gas inventories (NGHGIs) for Annex I (AI) countries of the United Nations Framework Convention on Climate Change (UNFCCC). Finally, the temporal relative change rates in ISA-driven carbon emissions were attributed to different driving factors, with contrasting patterns of emissions dynamics being identified for AI and NAI (i.e., non-Annex I) country groups, respectively.

### Annual ISA expansions and their land-cover sources

Four state-of-the-art global coverage remote-sensing products of impervious surface were used in our analysis. These included three products with a 30-m resolution (GAIA[6], GAUD[1] and GISA[17]) and the 300-m resolution ESA CCI land cover product[40] (for details refer to Supplementary Table 2). These products show comparable cumulative ISA expansions and their spatial distributions during the overlapping period of 1996−2015 (Supplementary Fig. 2).

We deemed that the mapped ISA or urban pixels in the four ISA products are highly dominated by ISA and are therefore considered to have 100% ISA coverage, irrespective of the naming of the land-cover type concerned (i.e., 'impervious surface' or 'urban') (see Supplementary Information 1.1 for the detailed justification for each product). This approach ignores any potential sub-pixel green areas within the mapped ISA or urban pixels and is consistent with the landscape homogeneity assumption underlying land cover classification and its spatial mapping (Supplementary Information 1.1). This imperfect but pragmatic approach has been widely applied in previous studies investigating global and regional ISA expansion[4,13,14]. We argue that, despite sub-pixel vegetated areas being ignored, such an approach will

unlikely result in any overestimation of global ISA expansion. First, the ignored potential urban green areas within ISA pixels are supposed to be offset by the omitted sub-pixel ISA areas within pixels identified as vegetated land. An analysis of three example cities shows that the ISA derived by the products used in this study was comparable to that derived using remote sensing with a very high spatial resolution (0.1 or 10 m) which could be considered as ground truth (Supplementary Information 1.2). Second, a rough estimate conducted by the authors shows that global minor roads might have accounted for 12.6–18.4% of the global ISA expansion but were likely omitted by the four products used in this study (Supplementary Information 1.3).

The 30-m resolution products were primarily used because they better support the landscape homogeneity assumption mentioned above and capture more accurately the spatial patterns of ISA than the MODIS-based 500-m product[1,6,23]. The three 30-m resolution products were selected in this study (out of several available 30-m resolution products) because they covered a period longer than 30 years with annual temporal resolution (Supplementary Table 2), which allows exploration of the long-term temporal dynamics of ISA expansion and the associated carbon emissions. The 300-m ESA CCI urban product covering 1992–2020 was also used, despite its medium resolution, because it provides a similar amount of ISA expansion, with similar spatial patterns, as the three high-resolution products (Supplementary Fig. 2), and because the ESA CCI land cover product was used to identify the source land cover for ISA expansion (detailed below).

ISA expansion was defined as transitions from a non-ISA pixel to an ISA one between two consecutive years. All ISA expansions were quantified as area fractions at 300-m resolution to facilitate the identification of source land cover using the ESA CCI land cover product. Note that, in GAIA, GAUD and GISA, the only information provided was the year and the location at which the concerned 30-m pixels were identified as ISA, and as a result, with these datasets, it is not possible to identify ISA pixels that revert back to non-ISA ones. For ESA CCI, occasional ISA to non-ISA pixel transitions did occur, but the total area undergoing such a phenomenon of 'urban green recovery' accounted for only 0.02% of the total area of ISA expansion over 1992–2020. In addition, a recent study reported that global areas undergoing 'urban green recovery' were two orders of magnitude smaller than the area of urban expansion over 1985–2015[1]. These results show that, over the past three to four decades, global urban dynamics have been dominated by ISA expansion while transitions from ISA to vegetated land have been negligible. Such a monotonous flow from vegetated land to ISA, supported by remote sensing evidence and driven by continuous growth in global urban population, has also been adopted in future projections of urban area dynamics (e.g., ref. 2). Therefore, here, we assumed that ISA expansions were irreversible and ignored transitions from ISA to vegetated land and their associated effects on terrestrial carbon dynamics.

Identifying the source land cover of annually expanding ISA areas necessitates overlaying the ISA products with a base land-cover distribution map. For this purpose, we chose the annual ESA CCI land cover products. There were two reasons for this choice: (1) no consistent, widely used 30-m resolution land cover products with an annual time step were available; (2) the ESA CCI land cover product has been widely used in previous studies to diagnose the source land cover information of urban or ISA expansion (e.g., ref. 1,2); the use of this product thus ensured consistency with previous analyses. For this purpose, non-ISA land cover types in the ESA CCI product were grouped into six types: forest, shrubland, wetland, grassland, cropland, and 'others' (including lichens and mosses, sparse vegetation, bare areas, water bodies and permanent snow and ice). Note that, here, the 'wetland' land-cover type was adopted directly from ESA CCI terminology. Forests in flooded areas with saline, fresh, or brackish water (e.g., mangrove), which are sometimes considered to be wetland, were included in the 'forest' land cover type in this study.

The source land cover of annually resolved ISA expansions according to each ISA product was diagnosed using the ESA CCI land cover distribution of the previous year. Because of the mismatch in the spatial resolution between ESA CCI product (300 m) and the high-resolution ISA products (30 m), and because of the inevitable local-scale inconsistency in land cover classification, there are cases where ISA expansion identified using the high-resolution products occurred over a 300-m pixel already classified as 'urban' by ESA CCI. In reality, this is possible because the sub-pixel vegetated area (which we ignored) within a 300-m pixel could be converted to ISA. The same cases were also reported in previous studies of ISA expansion, where this part of ISA transition was simply neglected[1]. However, we argue that this practice will underestimate the ISA expansion given by the high-resolution products. Instead, we followed the geographical similarity principle[41] to redistribute the 30-m scale ISA expansions over 300-m ESA CCI 'urban' land to different non-ISA source land on a 5-km grid, using the percentages of different non-ISA land cover types contributing to ISA expansion on the same 5-km grid. Note that this issue could only have been avoided if each ISA product included its own full set of land cover classification at an annual time step, which is unfortunately not the case. The same issue would also occur, even if some (high-resolution) land cover products, other than ESA CCI, had been used to identify ISA source land cover, due to inconsistencies in mapping at a local scale. But to our knowledge, no widely used annual 30-m land cover products are available for this purpose. Hence, we argue that the application of the geographical similarity principle is an imperfect but practical solution here.

The study period was finally determined as 1993–2018 based on two considerations regarding data availability: (1) the source land cover information of annual ISA expansions prior to 1993 could not be reliably identified because the ESA CCI product started in 1992; (2) to enhance our confidence on annual ISA expansions, we ensured that each year of our study period was covered by at least three ISA products.

## Carbon losses from live biomass, surface litter and dead wood due to ISA expansion

Carbon contained in the live biomass (both above- and below-ground), as well as in surface litter and dead wood, was assumed to be released into the atmosphere within the same year that vegetated land is converted to ISA. This approach is consistent with the Tier 1 Approach laid out in the IPCC guidelines on quantifying carbon losses in these carbon pools upon land conversion to settlements[42]. The same approach was also used in recent studies quantifying emissions due to forest loss[21]. Here, we included biomass carbon losses from the conversion of forest, shrubland, wetland, cropland, and grassland to ISA. The land cover type of 'others' was regarded as having ephemeral live biomass or surface litter, so its biomass carbon losses could be fairly ignored. Biomass carbon emissions were thus estimated as long as the identified ISA expansion was co-located with a 300-m ESA CCI pixel with a land-cover type of forest, shrubland, wetland, cropland, or grassland. For those ISA expansions derived from 30-m high-resolution ISA products occurring over 300-m ESA CCI 'urban' pixels, the associated biomass carbon emissions were estimated using the redistributed ISA source lands from the five vegetated land cover types on the 5-km grid (described above in detail), along with the average biomass carbon densities of the respective land cover types.

We used three static global biomass density maps: the 'Spawn' map[20], the 'CCI' map[22], and the 'Harris' map[21]. The Spawn map was developed recently by harmonizing various vegetation-specific above-ground (AGB) and below-ground (BGB) live biomass density maps with spatial resolutions as fine as 25 m. The dataset provides live AGB and BGB carbon density maps at a 300-m spatial resolution circa 2010 for different vegetation types. The reasons to use this dataset include: (1) it is based on the land cover map of ESA CCI, the same land cover map

used to derive source land types for ISA expansion, and integrates multiple biomass maps for woody vegetation, tundra vegetation, grassland and cropland; (2) living trees are defined as those with diameter at breast heights (DBH) larger than 10 cm and any amount of tree cover is included once detected using fine-resolution (30 m) tree cover maps[43], making this dataset capable of representing biomass carbon density for regions with sparse tree cover, which are often omitted in existing biomass density maps; (3) the integration of woody AGB maps specifically for Africa[44] makes it capable of representing biomass carbon density for savannahs and shrublands, which are typically ignored in existing biomass maps focusing on forest biomes.

The CCI biomass product was released by the ESA CCI BIOMASS project. Its most up-to-date version, 4.0, provides forest AGB density maps for the years 2010, 2017, 2018, 2019 and 2020 with a spatial resolution of 100 m[22]. The product was generated by integrating multiple observations, including ESA's C-band, JAXA's L-band Synthetic Aperture Radar and spaceborne LIDAR, and using AGB retrieval algorithms of improved allometries. The forest AGB map for 2010 was used, for two reasons: (1) it shows biomass density distribution for a time period close to that of the Spawn map[20]; and (2) it is less influenced by ISA expansions than other maps of more recent years (i.e., 2017, 2018, 2019 or 2020). The use of temporally changing biomass density maps was also tested as an alternative approach, but finally, we deemed using static maps to be more appropriate (detailed below).

The Harris map is a woody live AGB map, circa 2000, with a spatial resolution of 30 m[21]. This map combines a global forest AGB map and another AGB map specifically improved for mangroves, both of which were derived by combining height-biomass equations with remote sensing observations of canopy height from airborne or spaceborne lidar and large numbers of ground biomass measurements. The Harris map was selected because it has previously been used to quantify terrestrial carbon losses from forest loss[21] and because it is for an early time period, circa 2000, which means that it is not influenced by ISA conversion after 2000.

We also explored the possibility of using annually resolved dynamic biomass density maps by combining the Spawn map with two dynamic biomass density maps[22,45]. The results show that the differences in estimated woody biomass carbon emissions from ISA expansion among the different static maps were far greater than the difference made by the choice of either a static or a dynamic biomass density map (Supplementary Fig. 11). This implies that the uncertainty in the estimated ISA-driven woody biomass losses is influenced more by different biomass density products than by using either static or dynamic maps. In addition, using static biomass density maps was found to be state of the art for studies looking to quantify terrestrial biomass carbon stock changes induced by land cover change. Such studies typically combine static biomass density maps with land cover change maps (e.g., refs. 21,45–50). For this reason, the average of the Spawn, Harris and CCI woody biomass products (all re-sampled to 300 m) was used to quantify ISA-driven woody biomass losses in order to reduce the uncertainty that would be caused by using a single product. Over non-woody vegetation, the single Spawn map was used to derive biomass carbon emissions from ISA expansion, due to data availability. The average of the three AGB maps for forest and shrubland, and the Spawn AGB map for cropland, grassland and wetland were combined to form a global AGB map, which was then converted to the AGB carbon (AGBC) map.

Potential sub-pixel ISA within 300-m ESA CCI vegetated pixels (supposed to offset the sub-pixel vegetation area within ISA pixels that is omitted in this study) can render their observed biomass densities invalid (see Supplementary Information 2). Therefore, the ISA fraction within 300-m vegetated ESA CCI pixels was calculated based on the 30-m ISA products. Biomass densities for 300-m vegetated pixels with an existing ISA fraction >5% were interpolated using the 20 nearest valid biomass density observations (i.e., 300-m vegetation pixels with the

existing ISA fraction <5%) with the same land cover type using the inverse distance weighting algorithm[51]. The error in the estimated globally ISA-driven emissions due to biomass density interpolation was shown to be negligible. For the detailed reasoning underlying the interpolation of biomass density and the quantification of the associated uncertainties in the global ISA-driven emissions, please refer to Supplementary Information 2.

The root-to-shoot ratio of BGBC (BGB carbon) to AGBC (AGB carbon) from the Spawn map (which was also interpolated using the same method used to interpolate the combined AGBC map) was used to estimate the ISA-driven below-ground live biomass emissions. Surface litter and forest or shrubland dead wood carbon densities were estimated by assuming that they are constant fractions of AGBC density following the approach used in ref. 21. These fractions depend on global ecological zones, elevation and annual precipitation and were retrieved from a look-up table provided by Harris et al.[21].

## Carbon loss from soil organic carbon due to ISA expansion

Unlike carbon losses from biomass, for which only transitions from forest, shrubland, wetland, cropland, and grassland to ISA were accounted for, soil organic carbon (SOC) losses from ISA expansions on all source land covers were included. For ISA expansions identified by 30-m ISA products occurring over 300-m ESA CCI 'urban' pixels, the SOC emissions were estimated using the same method as used to estimate the corresponding biomass carbon losses (detailed above), but unlike biomass carbon loss, SOC losses due to ISA expansion from the land cover type of 'Others' were specifically included.

A gridded SOC stock dataset, the spatial distribution of ISA expansion, and the percentage of SOC loss following ISA establishment were combined to derive ISA-driven SOC losses. The quantified SOC losses were hence committed emissions. SOC losses were estimated for a soil depth of 0–30 cm, consistent with the default soil depth applied in the IPCC guidelines for estimating changes in SOC stock in NGHGIs[42]. The SoilGrids250m v2.0 dataset[19] was used to obtain the global distribution of 0–30 cm SOC density following a resampling of the original 250-m resolution data to 300-m resolution. Existing ISA within the pixels of the SoilGrids250m dataset might render the SOC density to be underestimated. To exclude such effects, rather than using the original SOC densities reported by Soil-Grids250m, SOC densities for all 300-m vegetated pixels of ESA CCI, but containing >5% ISA according to 30-m ISA products, were interpolated using the 20 nearest valid SOC densities (i.e., 300-m pixels with ISA fraction <5%) with the same land-cover type. The errors in the global ISA-driven SOC emissions due to spatial interpolation were estimated to be negligible (Supplementary Information 2).

To represent the upper and lower estimates of 0–30 cm SOC loss following ISA conversion, two constant SOC loss ratios were applied: 59.5%, as obtained by a literature synthesis as part of this study (derivation detailed below) and 20%, as suggested in the Tier 1 Approach in the IPCC guidelines[42]. The justification for employing the upper and lower estimations of ISA-driven SOC loss is detailed below.

The observed difference between SOC stock under ISA and in urban green areas from paired-site measurements was used to approximate SOC loss following ISA establishment on originally vegetated land because direct observations on the latter are rare. The online literature was searched using ISI Web of Science, Google Scholar, and the China National Knowledge Infrastructure (CNKI) using the keywords "impervious surface", "pavement", "sealed ground", or "built-up". Publications fulfilling the following criteria were included: (1) field sampling was conducted under both ISA and urban green areas; (2) SOC stocks were provided or could be estimated from the reported soil organic carbon concentration, bulk density and soil depth.

The search yielded a total of 14 studies, covering 6 countries, most of which were published between 2017 and 2021 (Supplementary

Table 3). The following information was extracted from each study: source, site location, SOC under ISA, SOC under neighboring urban green areas, the ISA type (e.g., road, building, etc.), soil sampling depth, age since conversion to ISA (if available) and soil physical properties (if available). It is noteworthy that we excluded a study by Ding et al.[52] from our analysis because their paired-site observations contain some aberrant measurements with SOC under ISA being greater than those under adjacent green areas, which was not reported by any other studies we collected. The final dataset includes 22 observations for soil depths up to 100 cm compiled from 19 paired sites (Supplementary Fig. 12). The raw data were obtained either from tables or from figures using the GetData Graph Digitizer (http://getdata-graph-digitizer.com/). For those studies providing only SOC concentration, the following equation was used to convert SOC concentration to SOC stock:

$$C_s = \frac{C_C \times BD \times D}{100} \tag{1}$$

where $C_s$ is SOC stock (kg C m$^{-2}$); $C_C$ is SOC concentration (g kg$^{-1}$); BD is soil bulk density (g cm$^{-3}$); and D is soil thickness (cm). The percentage of SOC loss under ISA was approximated by comparing carbon density between ISA and urban green areas as:

$$l_{ISA} = (1 - C_{ISA}/C_{GA}) \times 100 \tag{2}$$

where $l_{ISA}$ is the percentage of SOC loss due to conversion of vegetated land to ISA (%); $C_{ISA}$ is SOC density under ISA; $C_{GA}$ is SOC density under urban green areas. Our analysis shows that the SOC loss percentage has no relationship with soil depth ($n = 22$, $p > 0.05$). Due to the paucity of field observations, we were unable to investigate the relationships between SOC loss percentage and environmental factors. Hence the average percentage of SOC loss of all the observations for different depths was obtained and used to represent that for the 0–30 cm soil layer.

Eventually, we found that $59.5 \pm 16.6\%$ of SOC was lost following ISA establishment (Supplementary Fig. 12). On the one hand, the limited number of observations ($n = 22$) prevented explicit spatial mapping of the SOC loss ratio using complex statistical methods such as machine-learning models. But, on the other hand, almost all the studies in this synthesis were published within the last four years, suggesting that SOC dynamics following land conversion to ISA is an emerging topic in the field of global carbon-cycling research which is receiving growing attention.

To verify the validity of the SOC loss ratio derived from our small sample size, we further compared SOC loss under ISA with that under long-term bare fallow, as, in both cases, carbon inputs into the soil from plant photosynthesis are completely suppressed, either by land surface sealing or active vegetation removal. Changes in SOC should thus be dominated by the microbial decomposition of the previously remaining soil organic matter. To achieve this comparison, we synthesized the findings from Barré et al.[53], who reported the results of monitoring SOC stocks for over 30 years at seven long-term bare-fallow experiments in six sites across five countries. Detailed information, including site location, former land cover type, monitoring duration, sampling depth, and initial and final soil bulk densities, is provided in Supplementary Table 4. Barré et al.[53] also reported the final stabilized SOC concentration under the presumed condition of ever-lasting fallow using regression models. The SOC concentrations at the first and the last sampling time, as well as the final stabilized value, were converted to SOC stocks using Eq. (1). The bulk density when the SOC concentration reaches a stable value was assumed to be equal to that at the last sampling. The percentages of SOC loss due to bare fallow were estimated for both the last sampling time ($BF_{obs}$) and the time at which SOC finally reaches a stable level ($BF_{final}$) (Supplementary Fig. 12).

The synthesized data from bare-fallow experiments indicates that the percentage of SOC loss when the last samples were taken over bare-fallow sites ($BF_{obs}$), i.e., corresponding to a mean fallow age of 46 years, was 39.1% on average, significantly lower than the reported loss under ISA ($p < 0.05$). However, when SOC was predicted to finally stabilize under bare fallow at an age of around 80 years, the SOC loss ratio ($BF_{final}$) would reach 74.7%, significantly higher than the SOC loss under ISA ($p < 0.05$). The observation that the SOC loss following ISA establishment falls between the observed and finally stabilized SOC loss under permanent bare fallow gives credibility to these data.

The Tier 1 Approach in the IPCC guidelines suggests that for land converted to paved settlements (i.e., impervious surface areas), 20% of the soil carbon relative to the previous land use should be considered as lost[42]. However, this suggested loss ratio lacks any justification or support from site-level observations. Out of the 22 SOC loss ratio observations collected in this study, only one (17.2%), is close to the IPCC default value, with all the others being higher than the default value (Supplementary Fig. 12). In addition, a nationwide study of China incorporating 148 paired measurements of SOC for depths up to 100 cm on adjacent sites of ISA and pervious areas (i.e., lawns, bare land, etc.) reported that, on average, SOC under ISA was 38% lower than the value in pervious areas[52]. The reported field-based SOC loss ratios of 38% for China and 59.5% across the northern hemisphere are highly credible, as they are close to the SOC loss ratios derived from permanent bare-fallow experiments, which share a similar mechanism to that underlying SOC loss due to ISA expansion. Nevertheless, both our synthesis of studies and the nationwide study of China focus on the northern hemisphere. Using the mean SOC loss ratio of 59.5% derived from our synthesis will have a strong northern hemisphere bias, but using the IPCC default value of 20% will almost certainly underestimate SOC loss following ISA expansion. Hence, we adopted the compromise of using the IPCC default value as the lower boundary, to be comparable with the value used in NGHGIs, while our derived value was employed as the upper boundary to account for the observations.

## Comparison with carbon emissions from settlement expansion for Annex I countries of UNFCCC as reported in NGHGIs

The estimated ISA-driven emissions were compared with carbon sinks and/or sources from land conversion to settlements for Annex I countries of UNFCCC for 1993–2018, based on the national greenhouse gas inventories (NGHGIs) submitted to UNFCCC. Annex I countries are obliged to submit the common reporting format (CRF) tables along with the national inventory report (NIR) on an annual basis. The land conversion to settlements figures include forest converted to settlements (FS), cropland converted to settlements (CS), grassland converted to settlements (GS), wetland converted to settlements (WS), and other land converted to settlements (OS). Information on both land area converted to settlements and existing areas of settlements, as well as net carbon stock changes for each category, is provided in CRF tables available for 1993–2018. Net carbon stock changes include changes in live biomass, surface litter, dead wood, and the top 30 cm of mineral soil. For live biomass carbon stock, gross carbon gains, and gross carbon losses are additionally provided.

The "settlements" land use category used by NGHGIs has a broader scope than the impervious surface area in this study. According to the IPCC guidelines, "settlements" includes "all developed land, i.e., residential, transportation, commercial, and production (commercial, manufacturing) infrastructure of any size, unless it is already included under other land-use categories"[15]. Hence, it includes not only impervious areas but also herbaceous perennial vegetation such as turf grass and garden plants, trees in rural settlements,

homestead gardens and urban forests. Consequently, the settlement area is much larger than the ISA obtained in this study, particularly for the USA (Supplementary Fig. 5).

However, we expect that carbon sources from settlement expansion reported in NGHGIs should, in principle, be dominated by converting non-settlement vegetated land to the impervious land in the "settlements" category rather than to vegetation that is considered to belong to "settlements" by Annex I countries. This expectation was partly confirmed by the national inventories of Annex I countries showing a negligible carbon gain compared to carbon losses in biomass (Supplementary Fig. 1). Based on this, our estimated ISA-driven emissions were compared with carbon sinks and/or sources from land conversion to settlements for Annex I countries. Another factor to be considered when comparing the ISA-driven carbon emissions derived in this study with those in the NGHGIs was that the NGHGIs assume the loss in SOC to take place over a period of 20 years since settlement establishment, whereas, here, the SOC loss was quantified as committed emissions. For carbon losses in biomass, surface litter, and dead wood, however, both approaches assumed immediate loss. Nonetheless, neither the ISA-driven SOC emissions reported in this study nor those in the NGHGIs showed a temporal trend over 1993–2018 for Annex I countries (both with $p > 0.05$). Given that, in the absence of a temporal trend, the annually resolved emissions are the same as committed emissions, the lack of trend in both estimates enables their comparison.

## Socioeconomic drivers of ISA-driven terrestrial carbon emissions

We developed an "ISA-driven Emissions Identity" approach to attribute changes in ISA-driven terrestrial carbon emissions over 1993–2018 to different driving factors. This approach is analogous to the Kaya Identity used in fossil fuel $CO_2$ emission attribution studies[25]. According to our approach, annual carbon emissions ($E$) are expressed as the product of total **p**opulation ($P$), **u**rbanization rate ($u$, the ratio of urban population ($Pu$) to $P$), **r**esidential ISA intensity ($r$, the ratio of the existing $ISA$ to $Pu$), ISA expansion **s**peed-up factor ($s$, the ratio of annually expanding ISA area ($\Delta ISA$) to $ISA$), and the carbon **e**mission intensity ($e$, the ratio of $E$ to $\Delta ISA$) as follows:

$$E = P \cdot \frac{Pu}{P} \cdot \frac{ISA}{Pu} \cdot \frac{\Delta ISA}{ISA} \cdot \frac{E}{\Delta ISA} = Purse \qquad (3)$$

The relative proportional change rate of each variable $X(t)$ ($X$ refers to $E$, $P$, $u$, $r$, $s$, and $e$) is defined as $r(X) = (dX/dt)/X$. Applying a natural logarithm transformation to both sides of Eq. (3) and then taking differentials for each term, gives $(dE/dt)/E = (dP/dt)/P + (du/dt)/u + (dr/dt)/r + (ds/dt)/s + (de/dt)/e$, i.e., the relative change rate in $E$ ($r(E)$, in % per year) can be expressed as the sum of the relative change rates of all the driving factors:

$$r(E) = r(P) + r(u) + r(r) + r(s) + r(e) \qquad (4)$$

Based on the observed annual time series of $E$, $P$, $u$, $r$, $s$ and $e$, the relative change rate for each variable was obtained as the slope derived from a linear regression with time divided by the mean value over a given period, all being expressed in % per year.

The selection of underlying factors was based on empirical knowledge of the dynamics of urban expansion and urban-related carbon emissions as revealed by previous studies[54–56]. The total population ($P$) is a component of the original Kaya identity method[24] and is a fundamental driver for human socioeconomic development. The urbanization rate ($u = Pu/P$) reflects the population structure underlying ISA expansion[54]. Both population ($P$) and the urbanization rate ($u$) have been used as key variables to construct the narratives of the Shared Socioeconomic Pathways (SSPs) used to project future

anthropogenic $CO_2$ emissions in the most recent IPCC assessment report (AR6)[56,57], highlighting their fundamental roles in driving urban expansion and the associated carbon emissions. The residential ISA intensity ($r$) represents the ISA per urban capita and measures the 'affluence' level in terms of habitation space. Similar factors of urban area per capita have been used to decompose historical changes in urban energy $CO_2$ emissions by using a Kaya-like method[55]. The carbon emission intensity ($e$) represents the emissions per area of expanded ISA: an indicator with policy implications for land use decisions on ISA expansion. For instance, to reduce ISA-driven emissions, new ISA should be preferably allocated to land with a low carbon density for both biomass and SOC. The ISA expansion speed-up factor ($s$) is used to characterize the temporal dynamics of ISA expansion: $d(\Delta ISA/ISA)/dt = 0$ means that ISA has been expanding over time but with a constant relative change rate; $d(\Delta ISA/ISA)/dt > 0$ means the ISA growth rate has been speeding up over time; $d(\Delta ISA/ISA)/dt < 0$ means that ISA is still growing but the relative growth rate has declined over time.

Note that to apply this decomposition framework, the preconditions of $\Delta ISA > 0$ and consequently $E > 0$, are required to be met for all years of the period 1993–2018. These preconditions were verified to be fulfilled for the whole globe, and for both AI and non-AI countries.

In addition, variations in the relative change rates of $E$, as well as its underlying driving factors among different nations, were explored as a function of the GDP per capita ($GDP_{cap}$) during 1993–2018 to reveal connections between ISA-driven carbon emission dynamics and socioeconomic conditions. Note that as we omitted any cases of $\Delta ISA < 0$ (i.e., cases of ISA being converted back to vegetated land, see the explanations above), for most countries, we will have $\Delta ISA > 0$ for all years of 1993–2018 and rarely $\Delta ISA = 0$. Thus, we first selected countries with $\Delta ISA > 0$ for all years of the period 1993–2018 and with $GDP_{cap}$ data available for 1993. This selection consisted of 123 countries, whose total ISA-driven carbon emissions accounted for 85.2–86.0% of the total global emissions. These countries were then sorted according to ascending $GDP_{cap}$ and divided into 10 groups containing an equal number of countries. Finally, the median values of the relative change rates in $E$ and in the underlying driving factors for each country group were derived, and were fitted to a linear function of the median value of $GDP_{cap}$. Note that when performing this fitting, $GDP_{cap}$ was log-transformed, using 10 as the base, because the distribution of $GDP_{cap}$ among different nations is highly right-skewed (Supplementary Fig. 13). The results obtained by using $GDP_{cap}$ for the starting year of 1993 are presented in the main text. Repeating all three steps, but with a value of $GDP_{cap}$ for the end year of 2018 rather than for 1993, led to similar conclusions (Supplementary Fig. 10).

The conventional Kaya decomposition method has been reported to sometimes leave an unexplained residual in the relative change rate of the target variable (i.e., the value of "$r(E) - [r(P) + r(u) + r(r) + r(s) + r(e)]$" is not zero)[58,59]. In our case, the residuals for global, NAI countries, and AI countries were 0.09% $yr^{-1}$, 0.06% $yr^{-1}$, and 0.08% $yr^{-1}$, respectively, being an order of magnitude smaller than the relative change rate of all variables. As an alternative approach, we applied a complete decomposition approach, the logarithmic mean Divisia index (LMDI)[60] method, to validate the results derived using the Kaya method (for details, refer to Supplementary Information 3). The LMDI results indicate that the contributions of the underlying factors in ESA-driven emissions dynamics for global, NI and NAI countries are almost identical to those derived by using the Kaya-like method (Supplementary Fig. 14), suggesting a trivial role of the residuals in explaining the underlying drivers of ISA-driven emissions dynamics.

## Data availability

All datasets enabling our analysis are openly available. The GISA dataset is available at https://zenodo.org/records/6476661. The GAUD dataset is available at https://figshare.com/articles/dataset/

High_spatiotemporal_resolution_mapping_of_global_urban_change_ from_1985_to_2015/11513178/2. The GAIA dataset is available at http:// data.ess.tsinghua.edu.cn/gaia.html. The ESA CCI land cover maps are available at https://www.esa-landcover-cci.org/. The global above- ground and below-ground live biomass density products are available at https://doi.org/10.3334/ORNLDAAC/1763, https://data. globalforestwatch.org/datasets/gfw::aboveground-live-woody- biomass-density/about and https://data.ceda.ac.uk/neodc/esacci/ biomass/data/agb/maps/v4.0. The SoilGrids250m v2.0 dataset is available at https://files.isric.org/soilgrids. The datasets of GDP per capita, urban population, total population and the official Global Administrative Divisions are from the World Bank (https://data. worldbank.org/). Source data of SOC stock under ISA and urban green areas and SOC observations from the long-term bare-fallow experiments are provided with this paper.

## Code availability
Scripts used in this analysis are available through the corresponding author upon request.

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

## Acknowledgements

C.Y. is supported by the National Key Research and Development Program of China (Grant No. 2023YFB3907403), the Second Tibetan Plateau Scientific Expedition and Research Program (Grant No. 2022QZKK0101), and the Strategic Priority Research Program of the Chinese Academy of Sciences (Grant No. XDB40000000). C.Z. is supported by the National Natural Science Foundation of China (Grant No. 41861124003).

## Author contributions

C.Y. designed the research; L.Q. and J.H. performed the analysis; C.Y., L.Q. and J.H. wrote the draft. C.Y., L.Q., J.H., P.C., and C.Z. contributed to the interpretation of the results and to the writing of the paper.

## Competing interests

The authors declare no competing interests.
