## [Peer Review File · Nature Communications]

Substantial terrestrial carbon emissions from global expansion of impervious surface areaREVIEWER COMMENTS

Reviewer #1 (Remarks to the Author):

In recent decades, many studies explored the impact of urban expansion on the terrestrial vegetation carbon sequestration from local to global scales. For instance, Zhuang et al. (2023) demonstrates that global urban lands expanded by $37.60 \times 10^4 \text{ km}^2$ during 1990–2017 using 30m global impervious surface datasets, directly caused carbon loss by the destruction of vegetation (3.07 Pg C). In comparison, this manuscript quantified terrestrial carbon losses from global impervious surface area expansion over 1986–2020. In general, the content of this article is of significance, but the authors should compare their methodology and results with others and further discuss their findings and the novelty of this study.

Zhuang Q, Shao Z, Li D, et al. Impact of global urban expansion on the terrestrial vegetation carbon sequestration capacity. *Science of The Total Environment*, 2023, 879: 163074.

Other major concerns:

---This manuscript assesses carbon losses from various carbon pools including living biomass and soil organic carbon (SOC). However, Zhang et al. (2022) reported that the topsoils in the urban areas of cities with high levels of urbanization can act as carbon sinks due to the increase in vegetation. Urban green areas play important roles on the SOC stock in the topsoils. The contribution of SOC stock should be deeply discussed.

Zhang Z, Gao X, Zhang S, et al. Urban development enhances soil organic carbon storage through increasing urban vegetation. *Journal of Environmental Management*, 2022, 312(15): 114922.

---The study resampled the SOC-stock dataset from its original 250 m resolution to 300 m resolution, and then overlaid it with the maps of ISA expansions. It can cause a large uncertainty on the SOC-stock dataset and also the analysis on the impacts of global impervious surface expansion. Such uncertainty should be addressed.

---L47-50, L60-62: The manuscript raised the problems including the representations of ISA in DGVM, validation of ISA-driven carbon emissions, but it does not solve them. The introduction should be revised more concisely and clearly.

---L74-76: According to the manuscript: "our analysis focuses on the carbon effects of ISA expansion and excludes, conceptually, urban green areas (e.g., 75 street trees, residential gardens)". How to exclude the carbon effects of urban green areas (e.g., street trees, residential gardens)? The authors should clarify it by their four ISA data sources separately.

---L215: "the carbon emission intensity (e , the ratio of E to ΔU)". What does the carbon emission intensity (as the ratio of ISA-induced carbon emissions to annually expanding urban area) mean? What is the meaning of this indicator? According to the manuscript: "The relative change rate in emission intensity (e) showed no significant relationship with $\log_{10}(\text{GDPcap})$ " (L271-272). Should it be the ratio of ISA-expanded induced carbon emissions to annually expanding urban area?

---L212-216: ISA-induced carbon emissions, existing urban area, urban area expansion. The term "ISA" should be used for maintaining consistency with the topic of the manuscript.

---All the data sources and their meta data should be listed in a table for better readability and contrast.

Reviewer #2 (Remarks to the Author):

The authors of this manuscript have undertaken an ambitious task of tracing the global expansion of Impervious Surface Area (ISA) from 1986 to 2020, utilizing four distinct ISA or urban area products. They have attempted to quantify the associated carbon losses from biomass and topsoil by superimposing carbon density maps. The estimated results were subsequently juxtaposed with the carbon losses reported in National Greenhouse Gas Inventories (NGHGs) for Annex I countries. Furthermore, the authors have employed an "Urban Emission Identity" approach to attribute carbon losses to socioeconomic and urban dynamics factors.

While the scope of the study is commendable, it falls short of the expectations for a manuscript

intended for Nature Communications. The findings presented in this paper lack sufficient intrigue, and there is a pressing need for additional experiments to bolster the persuasiveness of the conclusions. Below are some specific comments and suggestions.

The accurate calculation of annual biomass carbon density is indeed crucial. In this study, land use/cover information and biomass carbon density data are derived from independent datasets. The authors need to clarify how they ensured the congruence of biomass carbon density with land use for years other than 2010. The potential data mismatch could introduce significant uncertainties.

Moreover, the study indicates that approximately 63% of ISA expansion occurred at the expense of cropland (Line 104-105). The biomass of cropland, as reported by Spawn et al. (2020, doi.org/10.1038/s41597-020-0444-4), could have significantly fluctuated between 1986 and 2020 due to factors such as climate change, advancements in agricultural technology, and intensification. While Supplementary Information 3 validates the results in the tropical forest region, the reliability of the estimated cropland biomass carbon loss warrants further verification. The data quality, particularly in relation to Soil Organic Carbon (SOC) and biomass carbon density, in areas transitioning from vegetated lands to ISA, is indeed a significant concern. While evaluations and validations have been conducted in previous studies that released these datasets, most of these assessments were performed in homogeneous vegetated areas. In contrast, areas near urban lands typically experience intensive land cover changes due to human activities, which pose substantial challenges for accurately estimating SOC and biomass carbon density. The carbon density data used in this study are typically located in these areas. Uncertainties originating from carbon-related data sources can indeed introduce significant uncertainties and biases into the carbon loss assessment in this study. Therefore, it is critical to address these uncertainties to ensure the study's findings are reliable. Furthermore, conducting a comparative analysis of results based on different biomass products could enhance the persuasiveness of the conclusions.

The application of the Kaya-like method for attributing changes in ISA-driven terrestrial carbon emissions to socio-economic driving factors does appear somewhat arbitrary. In Equation (3), the denominators can indeed be rearranged to form different driving factors, leading to varying conclusions. For instance, if we rearrange the equation as $E = P * P_u/U * U/\Delta U * \Delta U/P_u * E/P$, the driving factors could be interpreted as urban land per urban population, carbon emission per capita, and so on. Furthermore, it is feasible to incorporate other socio-economic variables into the equation, such as proportions of different economic sectors, electricity consumption, etc. This flexibility in the equation's formulation could potentially lead to a wide range of interpretations and conclusions. Therefore, it is crucial for the authors to justify their choice of variables and their placement in the equation to ensure the robustness and validity of their findings.

The authors have stated that the percentages of source land covers contributing to ISA expansion were derived from the ESA CCI product (Line 425-427). However, the ESA CCI product only covers the period from 1992 to 2020. This raises a valid question: how were the percentages of source land covers obtained for the years prior to 1992? This is a crucial detail that needs to be addressed to ensure the comprehensiveness and accuracy of the study. The authors should provide a clear explanation of their methodology for obtaining this data.

The acceleration rate of ISA expansion in this study, stated as $180.2 \text{ km}^2 \text{ yr}^{-2}$ (Line 110 and Figure 1-a), does seem puzzling. Both the studies releasing the GAIA and GAUD datasets report a steady growth of global ISA with no noticeable acceleration over the 30-year period. This discrepancy warrants further clarification.

The availability of live biomass carbon density maps only for the year 2010, and the lack of a time-series dataset for SOC density (SoilGrids250m), raises questions about how the authors derived the carbon density of transition lands during the study period from 1986 to 2020. This aspect needs to be addressed for a comprehensive understanding of the results and methodology.

The choice to treat the loss ratio of the IPCC default value as the lower boundary and the value derived from literature synthesis as the upper boundary for SOC loss is not immediately clear. The underlying logic behind this setup should be explained to ensure transparency in the methodology.

We sincerely thank both referees for their efforts in reviewing our paper and for their valuable comments that have helped us greatly improve the robustness of our findings and the quality of our manuscript. Please find below our point-by-point responses to the comments. To make it easy to track our revisions, all substantially modified text is marked in blue in the revised manuscript.

Reviewer #1 (Remarks to the Author):

[Comment 1] In recent decades, many studies explored the impact of urban expansion on the terrestrial vegetation carbon sequestration from local to global scales. For instance, Zhuang et al. (2023) demonstrates that global urban lands expanded by 37.60×10^4 km² during 1990–2017 using 30m global impervious surface datasets, directly caused carbon loss by the destruction of vegetation (3.07 Pg C). In comparison, this manuscript quantified terrestrial carbon losses from global impervious surface area expansion over 1986–2020. In general, the content of this article is of significance, but the authors should compare their methodology and results with others and further discuss their findings and the novelty of this study.

Zhuang Q, Shao Z, Li D, et al. Impact of global urban expansion on the terrestrial vegetation carbon sequestration capacity. *Science of The Total Environment*, 2023, 879: 163074.

[Response] We appreciate the time and effort that the referee has dedicated to providing valuable feedback on our manuscript and thank you for the constructive comments that have helped us improve the manuscript quality.

We thank the referee for referring us to Zhuang et al. (2023). Although both our study and Zhuang et al. examine the impacts of ISA expansion on processes related to the land carbon cycle, conceptually, the question addressed by our study fundamentally differs from that explored by Zhuang et al. (2023). Zhuang et al. (2023) quantify how much historical (1990–2017) ISA expansion within urban areas has reduced terrestrial net ecosystem productivity (NEP). As NEP measures how much carbon can be absorbed from the atmosphere (in a net amount sense) by an undisturbed ecosystem (Chapin et al., 2006), the cumulative loss of NEP quantified by Zhuang et al. (2023) actually measures the unrealized carbon absorption due to the replacement of plant ecosystems by ISA, but note that this unrealized carbon absorption adds no physically tangible CO₂ to the atmosphere — it is the amount of carbon that would be otherwise absorbed if the plant ecosystems had not been replaced by ISA. Likewise, Liu et al. (2019) quantify how much net primary productivity (NPP), which equates to plant gross photosynthesis minus plant autotrophic respiration) has been lost due to urban expansion but, again, this concerns the unrealized carbon uptake by plants (NPP is the sum of NEP and CO₂ release from the decomposition of surface litter and soil organic matter). In addition, the 3.07 Pg C of the cumulative loss of NEP (i.e., ‘unrealized carbon absorption’) due to ISA expansion within urban areas during 1990–2017 as reported by Zhuang et al. (2023) comes with a vague method description. According to their methods, this number seems to be calculated as the product of the change in NEP between 1990 and 2017, and the ISA area within the expanded urban boundary from 1990 to 2017.

In contrast, our study quantifies the amount of CO₂ released through replacing plant ecosystems by ISA. This CO₂ release arises from the decomposition of carbon-containing organic matter in plant biomass, surface litter and mineral soil of previous plant ecosystems that are now replaced

by plant-free ISA, and, unlike ‘unrealized carbon absorption’, it is physically tangible and contributes to the build-up of atmospheric CO₂ concentration. The loss of carbon contained originally in plant ecosystems prior to ISA establishment is well demonstrated by the clearing and loss of biomass (as is also reflected in IPCC guidelines), and by field experiments showing a much lower soil organic carbon stock in mineral soil under ISA than under the adjacent plant ecosystems (shown by our synthesis of global site-level measurements). These facts show that ISA expansion, as a form of land use change, is a source of anthropogenic carbon emissions that contributes to atmospheric CO₂ growth and the resulting climate change.

Given that the physical quantity estimated by Zhuang et al. (2023) is conceptually fundamentally different from that of our study, the numbers derived by the two studies cannot be simply compared.

Carbon losses due to ISA expansion have previously been quantified on national (e.g., Lai et al. (2016) for China for 1990–2010) or regional (e.g., Seto et al. (2012) for tropical regions for 2000–2030) scales. But, to the best of our knowledge, how global-scale ISA expansion has contributed to anthropogenic CO₂ emissions through a land-use effect, remains unknown and is neglected in the current global budget assessment. As explained in the Introduction of our manuscript, ISA expansion is a form of land use change (LUC), but the associated carbon emissions are largely ignored in either bookkeeping- or DGVM-based estimation of LUC emissions in the current global carbon budget assessments. Therefore, the key novelty of this study is that it quantifies carbon emissions from ISA expansion on the global scale and reports previously ignored but non-negligible contributions by ISA expansions to anthropogenic carbon emissions. In addition, we compare our estimation with that reported by national greenhouse gas inventories (NGHGs) for Annex I countries, and elucidate the underlying socioeconomic drivers for emissions dynamics.

We also thank the referee for referring to Zhuang et al. (2023) which points to a new ISA dataset of GISA (global impervious surface datasets), which we have included in our analysis in the revised manuscript. In view of the referee’s comments 5 and 7, we now use ‘ISA’ exclusively in the revised manuscript to restrict our research scope to ISA expansion by excluding any green areas within an urban environment. The scope of ISA is hence restricted to impervious surface areas which are artificial impermeable structures such as paved grounds, roads, built-up areas, etc. The second, but minor, difference between our study and Zhuang et al. (2023) is that our study covers all areas of ISA irrespective of whether they are in urban or rural environments (although ISA is very likely dominated by urban environments). Zhuang et al. (2023) focuses solely on ISA within urban areas by using an urban boundary map.

In response to the referee’s request, the novelty of our study in view of Zhuang et al. (2023) and similar studies are further clarified in the revised Introduction (lines 52–58).

Other major concerns:

[Comment 2] This manuscript assesses carbon losses from various carbon pools including living biomass and soil organic carbon (SOC). However, Zhang et al. (2022) reported that the topsoils in the urban areas of cities with high levels of urbanization can act as carbon sinks due to the increase in vegetation. Urban green areas play important roles on the SOC stock in the topsoils. The contribution of SOC stock should be deeply discussed.

Zhang Z , Gao X , Zhang S ,et al. Urban development enhances soil organic carbon storage through increasing urban vegetation. *Journal of Environmental Management*, 2022, 312(15).114922.

[Response] We agree with the referee that the effect of urban green space (or urban vegetation) on SOC stock dynamics is highly relevant for our study despite the fact that we focus on carbon effects of ISA which excludes large areas of urban vegetation. As reported by Zhang et al. (2022), urban green space has positive impacts on SOC accumulation especially in topsoils. On the one hand, land conversion from vegetation with relatively low SOC (e.g., cropland) to vegetation in urban green spaces with high SOC (e.g., lawn or urban forest) results in an increase in SOC. On the other hand, direct vegetation management in urban green spaces, such as irrigation, pest control and fertilizer application, and indirect anthropogenic effects, such as urban heat island effect and nutrient deposition, can lead to increased vegetation productivity, thereby increasing SOC by enhancing plant carbon inputs into soil. Similar findings to those of Zhang et al. (2022) at local and global scales have also been reported (e.g., Pouyat et al. (2009); Edmondson et al. (2014); Vasenev and Kuzyakov, (2018)), indicating that urban green spaces have greater rates of SOC accumulation than natural soils, and therefore soils in urban green spaces can act as potential carbon sinks within urban ecosystems.

However, although previous studies have provided a profusion of field observations on SOC stock in urban green spaces, challenges remain in upscaling these site-level observations to a larger regional or global scale (Guo et al., 2024). One of the critical challenges is the vagueness in the definition of urban environment or urban area. While a, more or less, common definition of ISA is agreed upon in the remote sensing community for the purpose of ISA mapping, and ISA is consistently used as a key feature to distinguish urban environment for urban area mapping, no commonly shared definition of 'urban area' exists within the academic community when the research objective goes beyond spatial mapping. For instance, a multitude of indicators have been employed to delineate urban area, including population density (Wu and Kim, 2021), built-up density (de Bellefon et al., 2021), nighttime light (NTL) density (Ch et al., 2021), and ISA fraction (Li et al., 2020). As a result, considerable uncertainties exist in the estimated urban area among different studies according to various definitions: the proportion of urban area to the world's total terrestrial area can range from 0.2% to 3% (Liu et al., 2014; Uchiyama and Mori, 2017).

Linked to the ambiguous definition of urban environment, the definition of urban green space (or urban vegetation) and its subsequent mapping remains a challenging task that has not yet been fully resolved. Due to varying objectives and methodologies across different disciplines, definitions of urban green space can vary (Taylor and Hochuli, 2017). For instance, Dallimer et al. (2011) investigated the impact of government policy on urban green space by including all vegetated areas within the administrative boundaries of 13 cities. Cummins and Fagg (2012) focused on the association between green space and weight status of urban residents and defined

urban green space as including parks, open spaces and agricultural land but not domestic gardens. Similarly, (Kabisch and Haase, 2013) examined the temporal changes of urban green space in the UK. They defined the urban green space as all vegetated areas larger than 25 ha situated within, or in contact with, urban fabric, but excluded city gardens, sports and leisure facilities (such as race courses, football stadiums, and tennis courts). These definitions of urban green space encompass land with both natural and man-made vegetation. Although both provide social and economic benefits, the vagueness in their definitions may lead to biased conclusions when quantifying their impacts on SOC.

Given the uncertainty in the area of urban vegetation, we chose to focus on quantifying terrestrial carbon losses from ISA expansion by assuming 100% coverage of ISA for the mapped ISA pixels with a given spatial resolution (see also our response to Comment 5). Another reason is that carbon balances of large tracts of urban vegetation identified as non-ISA, vegetated pixels in various land cover products have been largely accounted for in existing bookkeeping models or DGVMs in the current global carbon budget assessment, whereas the objective of this study is to account for the neglected carbon effects of ISA expansion which is missing from the carbon budget assessment.

In view of the referee's comment and previous findings that both vegetation productivity and SOC tend to increase over urban vegetation, we agree that if our study is extended to a broader scope of 'urban area', i.e., covering both ISA and urban vegetation, then part of the ISA-driven terrestrial losses could be offset by carbon sinks over urban vegetation. Unfortunately, to our knowledge, robust estimates on any substantial carbon sink in urban vegetation remain absent on a large spatial scale (partly due the uncertainty in the area of urban vegetation), making estimating the size of the offset effect highly challenging.

Following the referee's suggestion, in the revised Introduction (lines 81–86) and Discussion (lines 294–318), we have expanded the discussion of the carbon effects of urban green space and our justification for excluding urban green space.

[Comment 3] The study resampled the SOC-stock dataset from its original 250-m resolution to 300-m resolution, and then overlaid it with the maps of ISA expansions. It can cause a large uncertainty on the SOC-stock dataset and also the analysis on the impacts of global impervious surface expansion. Such uncertainty should be addressed.

[Response] It is a common practice to unify the spatial resolution of different datasets, often through resampling, when these datasets need to be used in a spatially consistent manner, including overlaying with each other (Sanderman et al., 2017; Harris et al., 2021). We chose 300 m as the primary resolution for the analysis in our study because (1) the ESA CCI dataset, which provides source land information for ISA expansion, has a spatial resolution of 300 m, (2) the biomass density datasets are either based on 300 m (Spawn et al., 2020) or could be readily re-scaled to 300 m because 300 m is an integer multiple of their original resolution (for CCI biomass product, the resolution is 100 m; for the Harris biomass product, the resolution is 30 m), (3) all high-resolution ISA products have a 30-m resolution which can be readily integrated to the 300-m resolution. Hence, we decided to resample the SOC density map from its original 250-m resolution to 300 m using the nearest neighbor method.

We agree with the referee that such resampling could lead to uncertainty in the estimated SOC losses due to ISA expansion (referred as ISA-driven SOC emissions). To estimate this uncertainty, we additionally used the SoilGrids250m SOC density map at its original 250-m resolution to calculate ISA-driven SOC losses and compared the results with those derived through SOC data resampling. The detailed procedures are:

- i) obtain the areas of annual ISA expansion for each 250-m pixel by directly upscaling from the original 30-m (i.e., GAUD, GAIA, and GISA) or 300-m (i.e., ESA CCI) ISA products;
- ii) interpolate the SOC densities for the 250-m pixels with an ISA fraction > 5% in 2018 by using SOC densities of the 20 nearest pixels with an ISA fraction <5% with the same source land cover type; (such interpolation results in negligible changes to the derived global SOC emissions, please refer to our response to Comment 4 raised by the #2 Referee. Likewise, interpolation of SOC density for 300-m pixels with an ISA fraction > 5% was also carried out when deriving SOC emissions through resampling SOC density data to a 300-m resolution).
- iii) compute ISA-driven SOC emissions at a 250-m spatial resolution by multiplying the ISA expansion area, the interpolated SOC densities and the SOC loss ratio;
- iv) compare the ISA-driven SOC emissions derived at 250-m and 300-m resolutions, at both temporal and spatial scales.

The results obtained by averaging the ISA-driven SOC emissions derived using the four ISA products (i.e., GAUD, GAIA, GISA and ESA CCI) are shown in Figures R1–R3. The area of the annual ISA expansions and ISA-driven SOC emissions were almost the same for the resolutions of 300 m and 250 m, for both temporal and latitudinal distributions. Based on these comparisons, we conclude that the resampling of the SOC density map resulted in little uncertainty and we therefore retained the practice of resampling the SOC density data from 250 m to 300 m in our study.

Figure R1 The annual expansion of global ISA (a) and the ISA-driven SOC emissions with the upper and lower boundaries of SOC loss (b) over the period of 1986–2020 derived using SOC density data with a spatial resolution of 250 m and by resampling the SOC data to 300-m resolution. The average of the four ISA products (i.e., GAUD, GAIA, ESA CCI and GISA) are shown.

Figure R2 The 35-yr total ISA-driven SOC emissions using the spatial resolutions of 250 m and 300 m with the upper boundary along the longitude (a) and latitude (b) with the interval of 0.5 degree. All the results shown here were the average of the four ISA products (i.e., GAUD, GAIA, ESA CCI and GISA).

Figure R3 The 35-yr total ISA-driven SOC emissions with the lower boundary along the longitude (a) and latitude (b) with the interval of 0.5 degree. All the results shown here were the average of the four ISA products (i.e., GAUD, GAIA, ESA CCI and GISA).

[Comment 4] L47-50, L60-62: The manuscript raised the problems including the representations of ISA in DGVM, validation of ISA-driven carbon emissions, but it do not solve them. The introduction should be revised more concisely and clearly.

[Response] We thank the referee for pointing out the potential confusion. The inconsistency in the representation of ISA (or urban area) in different DGVMs was not intended as a problem to be solved in this study, but was used as evidence to highlight the neglect of ISA-driven carbon emissions in the current global carbon budget assessments (i.e., a research gap this study intends to fill). This point has been further clarified in the revised manuscript as (lines 47–51): *“In addition, the representation of ISA (or urban land) is inconsistent among DGVMs, with it being either treated as pastureland or cropland, or bare land with no carbon fluxes (Supplementary Table 1). This neglect of carbon emissions from ISA expansion in both bookkeeping models and DGVMs demonstrates that these emissions are a missing component of the current global carbon budget assessment.”*

[Comment 5] L74-76: According to the manuscript: "our analysis focuses on the carbon effects of ISA expansion and excludes, conceptually, urban green areas (e.g., street trees, residential gardens)". How to exclude the carbon effects of urban green areas (e.g., street trees, residential gardens)? The authors should clarify it by their four ISA data sources separately.

[Response] We are grateful for the referee’s comment which helps us realize that the original sentence potentially leads to confusion. The paragraph concerned has been revised as (lines 81–89): *“Our analysis focuses on the carbon effects of ISA expansion by assuming 100% coverage of ISA for the mapped ISA pixels with a given spatial resolution, and any potential sub-pixel green areas (e.g., street trees, residential gardens) are thus ignored. Such an approach of assuming a 100% ISA coverage for the mapped ISA pixels is consistent with the landscape homogeneity assumption that underlies the classification of land cover and its subsequent spatial mapping (Supplementary Information 1). The same approach was also adopted in previous studies investigating global and regional ISA expansion and the associated effects on land carbon cycle^{4,13,14}. We argue that this is an imperfect but feasible approach that will unlikely lead to an overall overestimation of global ISA expansion, especially considering that ISA alongside global minor roads might have been omitted by these products (Supplementary Information 1).”*

Ignoring sub-pixel green areas is a deliberate choice. The underlying considerations that support this choice, and the revised paragraph given above, are now fully explained with detail in the revised Supplementary Information 1. In addition, following the referee’s comment, we have provided in Section 1.1 of Supplementary Information 1 the feasibility of the ‘100% ISA coverage’ assumption for each individual ISA product used in this study (lines 47–140 of Supplementary Information).

[Comment 6] L215: "the carbon emission intensity (e , the ratio of E to ΔU)". What does the carbon emission intensity (as the ratio of ISA-induced carbon emissions to annually expanding urban area) mean? What is the meaning of this indicator? According to the manuscript: "The relative change rate in emission intensity (e) showed no significant relationship with $\log_{10}(\text{GDP}_{\text{cap}})$ " (L271-272). Should it be the ratio of ISA-expanded induced carbon emissions to annually expanding urban area?

[Response] We are sorry for this confusion. The referee's understanding is correct. The carbon emission intensity (e , the ratio of E to ΔISA) is the ratio of terrestrial carbon emissions due to ISA expansions to annually expanding ISA. Its definition and meaning have now been further clarified in the revised manuscript as (lines 715–718): "*The carbon emission intensity (e) represents the emissions per area of expanded ISA: an indicator with policy implications for land use decisions on ISA expansion. For instance, to reduce ISA-driven emissions, new ISA should be preferably allocated to land with a low carbon density for both biomass and SOC.*"

Our revised analysis shows a declining emission intensity over 1993–2018 for both biomass and SOC which partly mitigates the carbon emission effect of acceleration in ISA expansion (for details, please refer to lines 237–247 in the revised manuscript). But again, the relative change rate in emission intensity over 1993–2018 shows no significant relationship with $\log_{10}(\text{GDP}_{\text{cap}})$ (i.e., the same conclusion as in the original manuscript). This was explained in the revised manuscript as (lines 278–281): "*The relative change rate in emission intensity (e) shows no significant relationship with $\log_{10}(\text{GDP}_{\text{cap}})$ (Fig. 5f), which is plausible because emission intensity largely depends on the carbon stock density of source land cover for ISA, which further depends on the natural conditions of different countries rather than on their economic development state.*"

[Comment 7] L212-216: ISA-induced carbon emissions, existing urban area, urban area expansion. The term "ISA" should be used for maintaining consistency with the topic of the manuscript.

[Response] We thank the referee for raising this important detail and apologize for the confusion introduced by using 'ISA' and 'urban' interchangeably in the original manuscript. Following the referee's suggestion, we now use 'ISA' or 'impervious surface area' exclusively throughout the revised manuscript because, conceptually, this study focuses on the carbon effects of ISA expansion only, and the influences of urban vegetated areas are not included (see also our responses to Comment 5). In addition, following the referee's Comment 2, the influences of omitting carbon balances in urban vegetation are discussed in the revised Discussion section (see our response to Comment 2).

[Comment 8] All the data sources and their meta data should be listed in a table for better readability and contrast.

[Response] We have summarized the data sources of paired-site SOC measurements under ISA and urban green space, and SOC observations from long-term bare-fallow experiments in a spreadsheet, and it is provided as the Source data with this manuscript. We have stated this information in Data availability section.

Reviewer #2 (Remarks to the Author):

[Comment 1] The authors of this manuscript have undertaken an ambitious task of tracing the global expansion of Impervious Surface Area (ISA) from 1986 to 2020, utilizing four distinct ISA or urban area products. They have attempted to quantify the associated carbon losses from biomass and topsoil by superimposing carbon density maps. The estimated results were subsequently juxtaposed with the carbon losses reported in National Greenhouse Gas Inventories (NGHGs) for Annex I countries. Furthermore, the authors have employed an “Urban Emission Identity” approach to attribute carbon losses to socioeconomic and urban dynamics factors.

While the scope of the study is commendable, it falls short of the expectations for a manuscript intended for Nature Communications. The findings presented in this paper lack sufficient intrigue, and there is a pressing need for additional experiments to bolster the persuasiveness of the conclusions. Below are some specific comments and suggestions.

[Response] We appreciate the referee’s efforts in reviewing our paper and the overall positive comments on our study. In particular, we highly appreciate the referee’s constructive comments that have helped greatly improve the manuscript quality.

Following the referee’s suggestions, additional experiments and substantial revisions have been made to enhance the robustness of our findings.

(1) The uncertainties in the derived biomass and SOC carbon losses due to ISA expansion, caused by interpolating for carbon densities for pixels with an existing ISA fraction > 5%, are now fully investigated.

(2) Dynamic biomass density maps are constructed based on the best available datasets and further used to derive ISA expansion-caused biomass carbon losses from the land cover types of woody land (forest and shrubland), grassland and cropland. However, likely due to data limitations, negligible differences are found between the results derived using dynamic versus static biomass density maps. Hence, static biomass density maps are still used in the revised manuscript. However, to reduce the uncertainty, we have used three different static woody biomass maps. Given that using static carbon density maps is the state-of-the-art in similar studies, and that this study is the first one to specifically target the terrestrial carbon losses due to ISA expansion, we argue that, although using static carbon density maps is sub-optimal, our study remains valuable in that it provides the first estimate of global-scale ISA-driven terrestrial carbon losses.

(3) The time period of our investigation has been modified to 1993–2018, in order to, a) remove the uncertainties caused by using the constant 1992 ESA CCI land cover product to derive source land cover information for annual ISA expansions before 1992 (1986–1992); and b) ensure that each year of the study period is covered by at least three ISA products. The shorter study period compared to our original manuscript does not change the conclusions but has helped to enhance data reliability and the robustness of our findings.

Please find below our detailed point-by-point responses to each comment.

[Comment 2] The accurate calculation of annual biomass carbon density is indeed crucial. In this study, land use/cover information and biomass carbon density data are derived from independent datasets. The authors need to clarify how they ensured the congruence of biomass carbon density with land use for years other than 2010. The potential data mismatch could introduce significant uncertainties.

[Response] Following the referee's suggestion, we examined the sensitivity of ISA-driven woody (i.e., forest and shrubland) biomass losses to using dynamic (annual) biomass density maps derived by using the best available remote sensing observations of woody biomass dynamics. We found that, although using dynamic biomass density maps slightly changed the estimated amount of ISA-driven woody biomass carbon losses compared to using a static map, the difference is much smaller than those derived by using different static biomass density maps. Hence the uncertainty in the estimated ISA-driven woody biomass carbon losses seems to be influenced more by the different biomass density products than by the choice of a static or dynamic map. For this reason, in the revised manuscript, the mean values of three static woody biomass density maps, in contrast to the single map used in our original analysis, were used to quantify ISA-driven woody biomass carbon losses (for applying dynamic biomass density maps for cropland and grassland conversion to ISA, please refer to below our response to Comment 3).

Additionally, we reviewed recent (after 2018) studies with a similar objective to ours (i.e., quantifying carbon losses due to land cover change) published in high-impact journals (Science, Nature, PNAS and their sub-journals) to check the usage of static or dynamic biomass density maps in these mainstream studies. We found an exclusive usage of static biomass density maps, clearly due to the limitation of data availability. We thus conclude that, despite potential uncertainties, using static biomass density maps seems to be the state-of-the-art. Given the novelty of our study, we argue that using static biomass density maps still provides valuable information on ISA-driven terrestrial carbon losses, although we completely agree with the referee that accurate dynamic biomass density maps remain highly desirable. Below, we provide evidence supporting our conclusions.

We searched the high-impact journals (Science, Nature, PNAS and their sub-journals) using the key words of "land cover change/conversion" or "forest loss" combined with "biomass carbon loss" or "biomass carbon stock change" to look for recent studies (after 2018) that used remote sensing-based datasets or methods to quantify terrestrial biomass carbon stock changes resulting from land cover change at the global or regional scales. Fourteen studies were identified (Appendix Table 1). These studies can be broadly classified into two categories in terms of the approach being used:

i) Annual land cover maps with a high (30-m) or medium (1-km) resolution were first used to derive land cover change (or forest gain and forest loss). Biomass density maps were then combined with information on land cover change to calculate biomass carbon stock changes (gains or losses). In this case, static biomass density maps, with a spatial resolution ranging from 30 m to 1 km, were used exclusively, due to the absence of long-term dynamic biomass density maps with a high or medium spatial resolution.

ii) Instead of first analyzing land cover change and then combining with biomass density datasets, remotely sensed vegetation index (such as EVI), or vegetation optical depth (VOD) was used to directly obtain annual time series of biomass maps. However, this type of remote sensing application typically has a coarse spatial resolution of 10 km (e.g., Xu et al. (2021)) or 25 km (e.g., Yang et al. (2023)).

Our study focuses on using annual land cover and ISA maps with a, mostly, high spatial resolution (30 m) to capture ISA expansions, including three 30-m ISA maps (i.e., GAUD, GAIA and GISA) and one 300-m land cover map (i.e., ESA CCI). ISA maps with 30-m or 300-m resolutions are necessary because the MODIS-based urban product with a coarser 500-m spatial resolution missed ~80% of ISA expansion (Sun et al., 2020). We hence deemed that high-resolution biomass density maps are needed in order to match the high-resolution land cover or ISA maps when estimating ISA-driven biomass carbon losses, making our study a typical study of the first category described above. Using static biomass density maps therefore seems justifiable, given that this practice is the state-of-the-art in similar studies and given that long-term, high-resolution dynamic biomass density maps are currently unavailable.

Nonetheless, we agree with the referee that annual biomass carbon density maps, rather than the static biomass carbon density map for a given year or period, will better match the annual land cover or ISA maps and help to more accurately estimate ISA-driven biomass carbon emissions. However, to the best of our knowledge, long-term global annual maps of biomass carbon density with high or medium spatial resolutions (ranging from 30 m to 1 km) are not yet readily available. We hence constructed them by integrating a medium-resolution (300 m) static biomass density map with coarse-resolution (10 km) annual biomass density maps and with high-resolution (100 m) dynamic biomass density maps, respectively.

In order to construct multi-year global biomass density maps, two global woody aboveground biomass (AGB) products covering multiple years and having a spatial resolution feasible for integration with the medium-resolution static biomass density maps were used: (1) the 10 km annual woody AGB maps covering 2000–2019 provided by Xu et al. (2021); (2) the 100 m woody AGB maps for 2010, 2017, 2018, 2019 and 2020 from ESA Climate Change Initiative (CCI) (Santoro and Cartus, 2023). Given that these two products overlapped for 2010–2019, we combined both with the static AGB map of Spawn et al. (2020) circa 2010 (referred as the Spawn static map) used in our original analysis to generate annual AGB maps for 2010–2019, respectively. Note that all these AGB maps were converted to AGB carbon (AGBC) maps with the unified unit of Mg C ha^{-1} .

More specifically, (1) for the annual AGB maps from Xu et al. (2021), changes in AGB between each year of 2011–2019 and 2010 were calculated at the same spatial resolution of 10 km. These AGB change maps were then resampled to 300 m using the nearest neighbor method and further applied on the Spawn static map to derive annual AGB maps with a 300-m resolution for 2010–2019. (2) For the AGB maps from ESA CCI, differences in AGB between each year of 2017–2019 and 2010 were computed at 100-m resolution and were then resampled to 300 m. The annual AGB change maps for 2011–2016 in comparison to 2010 were generated by linear interpolation between the years of 2010 and 2017. The annual AGB maps were then obtained by applying the annual AGB change maps on the Spawn static map for each year of 2011–2019. (3) To test the influence of the spatial resolution, we further resampled the 100-m annual AGB change maps for 2011–2019 from ESA CCI to 10 km, the same resolution as the AGB change maps of Xu et al. (2021). These 10-km AGB change maps from ESA CCI were then used to generate annual AGB maps following the same procedure as in Step (1). Afterwards, all three annual AGB maps were used to estimate ISA-driven woody biomass losses, following the same procedure as applied for using static biomass density maps, including necessary interpolation for AGB densities for 300m-pixels with

an ISA fraction > 5% (the error caused by interpolation is globally negligible, please refer to below our response to Comment 4).

Meanwhile, we found two other high-resolution static woody AGB maps feasible for our purpose when performing the literature review mentioned above. One is the forest AGB density map provided by Harris et al. (2021) for circa 2000 with a resolution of 30 m. The Harris map was selected for two reasons: (1) it has a high spatial resolution of 30 m; (2) it is for the period circa 2000, which means that this map is not influenced by ISA conversion after 2000. The other map is the 100-m forest AGB map for the year 2010 from ESA CCI. This map was selected for its high spatial resolution (100 m), and also because it shows biomass density distribution for a time period close to that of Spawn et al. (2020) (also circa 2010) and it is less influenced by recent ISA expansion than other years (2017, 2018, 2019 or 2020). Both static AGB maps are widely known in the community and have been used in studies quantifying terrestrial carbon changes caused by land cover change. They were also converted to AGBC maps with the unit of Mg C ha⁻¹.

The three annual AGBC density maps constructed above, along with the three static ones, were then utilized to estimate global ISA-driven woody biomass carbon emissions. Below-ground biomass carbon emissions were estimated using a static root-to-shoot ratio map derived by using BGBC (below-ground biomass carbon) and the AGBC map of Spawn et al. (2020). As is shown in Figure R4, the emissions estimated using all different maps show similar temporal variations, but the differences in emissions among the different static maps are much larger than those among the different dynamic maps. Compared to the result of the Spawn static map, the total biomass carbon losses over 2010–2019 derived using the ESA CCI static map are 16% higher, and those derived using the Harris map are 23% higher, whereas those derived by using the dynamic biomass density maps show differences of only -2%–2%. These results show that the uncertainty in the estimated ISA-driven woody biomass losses is influenced more by the different biomass density products than by the use of either static or dynamic maps.

However, we feel that using dynamic biomass density maps as constructed here might have underestimated their impacts because current medium-to-coarse-resolution (100 m to 10 km) remote sensing-based dynamic biomass datasets might underestimate highly local-scale biomass changes, such as forest biomass growth with age, gap-scale forest mortality or forest degradation caused by selective logging, given that the land cover type remains unchanged. For this reason, the use of dynamic biomass density maps might provide a false confidence of being precise but it cannot really reduce the uncertainty. This makes using static biomass density maps as both a practical choice and, as revealed from our literature review, the start-of-the-art for similar studies.

Given the data availability at the current stage, we consider that using static biomass density maps to estimate ISA-driven biomass emissions is feasible. In the revised manuscript, the average of three static woody biomass density maps, in contrast to a single map in our original analysis, were used to reduce the uncertainty in ISA-driven woody biomass losses (for applying dynamic biomass density maps for cropland and grassland, please refer to our response to Comment 3, below).

Following the referee's suggestion, we have updated the corresponding results of the ISA-driven woody biomass carbon emissions using the average of the three static biomass carbon density maps throughout the revised manuscript and the descriptions of the new static biomass density maps were added in the Methods (lines 479-525) and Supplementary Fig. 11.

Figure R4 ISA-driven woody biomass carbon emissions derived using the average ISA expansion of the four ISA products (i.e., GAUD, GAIA, GISA and ESA CCI) and different dynamic and static biomass density maps during 2010–2019. Dynamic_Xu indicates the emissions derived using the dynamic biomass density maps by integrating the static Spawn et al. (2020) map with the dynamic maps from Xu et al. (2021); Dynamic_ESA10 indicates the emissions derived using the dynamic biomass density maps by integrating the static Spawn et al. (2020) map with the dynamic maps from ESA CCI but first with resampling to 10 km. Dynamic_ESA300 is similar to Dynamic_ESA10 but with resampling to 300-m resolution. Static_Spawn, Static_Harris and Static_ESA2010 indicate the emissions derived using the static biomass density maps of Spawn et al. (2020), Harris et al. (2021) and 2010 ESA CCI map, respectively.

[Comment 3] Moreover, the study indicates that approximately 63% of ISA expansion occurred at the expense of cropland (Line 104-105). The biomass of cropland, as reported by Spawn et al. (2020, doi.org/10.1038/s41597-020-0444-4), could have significantly fluctuated between 1986 and 2020 due to factors such as climate change, advancements in agricultural technology, and intensification. While Supplementary Information 3 validates the results in the tropical forest region, the reliability of the estimated cropland biomass carbon loss warrants further verification.

[Response] The cropland biomass map of Spawn et al. (2020) integrates globally gridded information on crop yields for more than 70 annually harvested herbaceous commodity crops and remote sensing-based cropland annual net primary productivity (ANPP, which is equivalent to biomass carbon stock for annual crops) and is, to the best of our knowledge, the only widely used, medium-resolution cropland biomass data with a global coverage.

We agree with the referee that factors such as climate change, climate variation and advancements in agricultural technology have an influence on cropland biomass and therefore, dynamic (annual) cropland biomass maps are highly desirable for our analysis. In fact, this concern has also been shared by Spawn et al. (2020), whose cropland biomass map circa 2010 was generated by using the biomass map (based on globally gridded crop yield information) circa 2000 with the addition of ten times the Theil-Sen slopes of cropland ANPP during 2000–2015 as derived from the MODIS ANPP products.

As we are not aware of any widely used medium-resolution long-term cropland ANPP products other than MODIS, we followed an approach similar to Spawn et al. (2020) to derive annual cropland biomass maps over 2001–2020, using the static cropland biomass map from Spawn et al. (2020) (hereafter referred to as *Static_Spawn*) and MODIS cropland ANPP data. The MOD17A3HGFv061 ANPP product covering 2001–2020 with a spatial resolution of 500 m and annual time step was resampled to the 300-m resolution.

First, we reversed the process in Spawn et al. (2020) to derive the cropland biomass map for 2001 as the base year for our dynamic map construction. The Theil-Sen slopes of ANPP during 2001–2015 for 300-m grid cells of the cropland land cover type were calculated. The base map of cropland biomass circa 2001 was obtained by subtracting nine times the Theil-Sen slopes of ANPP for 2001–2015 from the benchmark (2010) map of *Static_Spawn* for cropland. Note that the base map circa 2001 rather than 2000, as in Spawn et al. (2020), was used because ANPP data for 2000 was not available from the MOD17A3HGFv061 product.

Second, two types of annual cropland biomass growth rate maps using MODIS ANPP data were derived, one using ANPP differences between every two consecutive years from 2001 to 2020, and the other using the Theil-Sen slopes over the period of 2001–2020. Two types of dynamic cropland biomass maps for 2001–2020 were then generated using the base map of cropland biomass circa 2001 plus the cumulative growth rates with the respective type for a given year, followed by spatial interpolation to derive biomass densities for 300-m pixels with an ISA fraction > 5% (the error caused by interpolation is globally negligible, please refer to below our response to Comment 4). The annual cropland biomass maps derived using the Theil-Sen slopes were referred to as *Dynamic_ANPPslope*, and the dynamic maps derived using the differences between two consecutive years were referred to as *Dynamic_ANPPdiff*. Note that all these biomass maps were converted to biomass carbon density maps with the unit of Mg C ha⁻¹.

The biomass carbon emissions due to ISA expansion over cropland for 2001–2020 were then estimated using dynamic cropland biomass carbon density maps of *Dynamic_ANPPslope* and *Dynamic_ANPPdiff*, as well as the static map of *Static_Spwan*. As shown in Figure R5, negligible differences were found in ISA-driven carbon losses between the two dynamic biomass carbon density maps and between either dynamic biomass carbon density maps and the static map.

Figure R5 Estimated biomass carbon emissions due to ISA expansion using the average of the four ISA products (i.e., GAUD, GAIA, GISA and ESA CCI) over cropland for 2001–2020 based on dynamic and static biomass carbon density maps. Dynamic_ANPPslope denotes the ISA-driven crop biomass carbon emissions using the dynamic biomass maps integrating the static biomass map from Spawn et al. (2020) with growth rates calculated from the Theil-Sen slopes of MODIS ANPP. Dynamic_ANPPdiff is similar, but with growth rates calculated from the ANPP differences between two consecutive years. Static_Spawn indicates the results of using the static crop biomass carbon density map from Spawn et al. (2020).

Given that grasslands and most croplands are herbaceous, we used the same methods as for cropland to derive annual grassland biomass carbon density maps and estimated biomass carbon emissions due to ISA expansion over grassland using both dynamic and static biomass carbon density maps. The results (Figure R6) again show negligible differences in estimated carbon emissions between using the dynamic and static biomass carbon density maps.

Figure R6 ISA-driven grassland biomass carbon emissions using the average of the four ISA products (i.e., GAUD, GAIA, GISA and ESA CCI) based on dynamic and static biomass carbon density maps for grassland during 2001–2020. Dynamic_ANPPslope indicates the ISA-driven grass biomass carbon emissions derived using the dynamic biomass maps integrated with the static biomass map from Spawn et al. (2020) with growth rates calculated from the Theil-Sen slopes of MODIS ANPP. Dynamic_ANPPdiff is similar, but with growth rates calculated from the ANPP differences between two consecutive years. Static_Spawn indicates the result of using the static grass biomass map from Spawn et al. (2020).

These results demonstrate that using the best available remote sensing-based datasets to construct dynamic biomass density maps does not substantially alter the derived biomass carbon losses from ISA expansion over cropland or grassland. Considering further that the constructed dynamic biomass density maps do not cover our entire study period (1993–2018), we have retained the static maps for both cropland and grassland in the revised manuscript.

[Comment 4] The data quality, particularly in relation to Soil Organic Carbon (SOC) and biomass carbon density, in areas transitioning from vegetated lands to ISA, is indeed a significant concern. While evaluations and validations have been conducted in previous studies that released these datasets, most of these assessments were performed in homogeneous vegetated areas. In contrast, areas near urban lands typically experience intensive land cover changes due to human activities, which pose substantial challenges for accurately estimating SOC and biomass carbon density. The carbon density data used in this study are typically located in these areas. Uncertainties originating from carbon-related data sources can indeed introduce significant uncertainties and biases into the carbon loss assessment in this study. Therefore, it is critical to address these uncertainties to ensure the study's findings are reliable. Furthermore, conducting a comparative analysis of results based on different biomass products could enhance the persuasiveness of the conclusions.

[Response] Following the referee's suggestion (made at the end of this comment), to reduce the uncertainty of the quantified carbon losses linked to carbon density maps, in the revised manuscript we used three different woody biomass density maps to obtain their mean values to quantify woody biomass losses due to ISA expansion. Please also refer to our response to Comment 2 above. However, due to data availability issues, we still use the single static biomass density map of Spawn et al. (2020) for cropland, grassland and wetland and the single static SOC density map of SoilGrids250m in the revised manuscript.

(1) land homogeneity assumption as the basis for carbon density interpolation

We agree with the referee that the carbon density maps used in this study are mainly validated for homogeneous landscapes. For example, the static biomass density map circa 2010 from Spawn et al. (2020) reports biomass densities for different land cover types using the land cover map distribution from the ESA CCI product of 2010. On the other hand, in our approach to quantifying ISA-driven carbon losses, carbon densities for pixels transitioning to ISA are primarily obtained from carbon density maps through identifying their land cover type information. Given that landscape homogeneity is the key assumption underlying the classification of land cover and its subsequent spatial mapping, we argue that obtaining carbon densities by combining carbon density maps and land cover maps is consistent with their underlying landscape homogeneity assumptions.

Nonetheless, we also agree with the referee that land cover changes due to human activities, as well as land management, are prevalent for land close to urban areas. In principle, land cover changes that are able to modify the landscape homogeneity at a scale of 300 meters will likely be captured by the annual ESA CCI land cover maps. Thus, using dynamic (annual) ESA CCI land cover maps to derive the source land cover of ISA expansion in this study will be able to largely account for this situation. Otherwise, if the considered land cover change occurs at a spatial scale far less than 300 meters (i.e. sub-pixel scale), annual ESA CCI land cover maps will not be able to detect clear land cover changes at the 300-m scale and will instead maintain the land homogeneity assumption (i.e., no land cover change being detected). In this case, however, carbon densities, in particular biomass carbon density, are expected to change even without land cover change at a 300-m resolution. We expect this effect will be captured by constructing dynamic biomass density maps. But according to our results (response to Comment 2), using dynamic biomass density maps to account for the effects of sub-pixel scale land cover change and land management leads to negligible differences in the derived ISA-driven biomass carbon losses. These results demonstrate

that, given the best currently available datasets of land cover and biomass density, using land cover information as the first-order determinant for biomass carbon density seems plausible.

However, in developing the methods of this study, our experiences revealed the fraction of ISA within a 300-m pixel as being an important factor breaking the land homogeneity assumption described above. Unlike vegetated areas, conceptually ISA (e.g., ground covered by tar, concrete, asphalt or mixtures) has almost zero biomass. Therefore, the fraction of ISA within a 300-m pixel can highly influence its biomass density even if the whole pixel could still be classified as a vegetated pixel according to the ESA CCI product. Hence, to strengthen the validity of the homogeneity assumption, we deemed all biomass observations for 300-m pixels with an existing ISA fraction exceeding 5% in 2018 (defined as ISA-contaminated pixels) as invalid. Carbon densities, including both biomass and SOC, for ISA-contaminated pixels, are interpolated using nearby ISA-free pixels, defined as pixels with an ISA fraction < 5% in 2018. The spatial interpolation was based on the assumption that, before any sub-pixel (300 m) scale ISA expansion according to 30-m ISA products, carbon densities for these pixels could be derived from neighbouring pixels of the same land cover type.

(2) Uncertainty caused by carbon density interpolation

Carbon density interpolation will introduce uncertainties to the estimated ISA-driven carbon emissions. As the (true) values of carbon densities for ISA-contaminated pixels (300-m pixels with >5% ISA fraction) under the land homogeneity assumption (i.e. before having any ISA) are unknown, the exact error caused by interpolation cannot be quantified. However, this error can be approximated by interpolating carbon densities for ISA-free pixels (300-m pixels with <5% ISA fraction) whose observed carbon densities (which can be considered as true values) are known and for which, therefore, no interpolation is needed. The detailed procedure for estimating the interpolation-caused uncertainty is as below:

i) identify all 300-m ISA-contaminated pixels whose carbon densities need to be interpolated. ISA-contaminated pixels are defined as those 300-m pixels where the total ISA area of the four ISA products (GAUD, GAIA, GISA and ESA CCI) exceeds 5% in 2018. In contrast, ISA-free pixels were identified as either those 300-m pixels without ISA or those for which the total ISA area of the four ISA products was smaller than 5% in 2018.

ii) Identify those ISA-free pixels around ISA-contaminated pixels, defined as buffer ISA-free pixels. The carbon densities of these buffer ISA-free pixels are known and are considered as true values.

iii) Interpolate the carbon densities for the above buffer ISA-free pixels using IDW (inverse distance interpolation) with carbon densities of the 20 nearest ISA-free pixels with the same land cover type.

iv) Calculate the relative error (RE) using the interpolated and observed (true) carbon densities for buffer ISA-free pixels, as below:

$$RE = \frac{Interpolated - Observed}{(Interpolated + Observed)/2} \times 100\%$$

This metric was selected for its symmetry so that the distribution of RE will not be skewed.

and v) Quantify the median and standard deviation of RE values for all the valid buffer ISA-free pixels at the global scale or for a given land cover type.

For biomass density interpolation, the average of the three biomass density maps (i.e., Spawn et al. (2020), Harris et al. (2020), and ESA CCI in 2010) for forest and shrubland, and the static biomass density map of Spawn et al. (2020) for cropland, grassland and wetland, were combined to form a global biomass density map. This combined static AGB map was utilized to estimate the interpolation error for biomass density. The spatial distribution of RE values on a 5-km scale is shown in Figure R7. Spatially, the RE values were rather heterogeneous. In terms of the distribution of RE, 73% of RE values were found within the range of -10% to 10%, and 94% were within -30% to 30%. At the global scale, the RE values, obtained by integrating all land cover types, is $1.3_{-2.0}^{7.1}$ % (median and interquartile range), suggesting a likely very small overestimation of ISA-driven biomass carbon emissions caused by carbon density interpolation. The RE for different land cover types range from $0.1_{-5.1}^{6.8}$ % for cropland to $3.3_{-1.1}^{11.3}$ % for forest (Figure R8).

Figure R7 Spatial distribution of RE (relative error, in %) for biomass density interpolation on a 5-km grid.

Figure R8 The RE (relative error) of biomass interpolation at the global scale and for different land cover types (a) and its histogram (b). The dots and the error bars in panel a indicate the median value and the interquartile range, respectively.

RE caused by SOC density interpolation shows a relatively homogeneous spatial distribution (Figure R9). In terms of the statistical distribution of RE, 98.7% of RE values are found within the range of -5% to 5%, and 99.8% are found within the range of -10% to 10% (Figure R10b). The global RE for SOC interpolation is $0.1_{-0.4}^{+0.6}\%$. RE is also very small across the different vegetation types (Figure R10a).

Although RE provides an approximate for the error in the interpolated carbon densities, the true value of error could not be known exactly. Hence, we avoid correcting the estimated ISA-driven emissions in 5-km grids by using the information of RE. Rather, the statistical distribution of RE was used to calculate the uncertainties of the derived ISA-driven biomass, SOC, and total carbon emissions. Since the ISA-driven carbon emissions are the products of carbon density maps and annual Δ ISA maps, given that Δ ISA is independent of the error in carbon density interpolation, RE values for carbon densities can then be directly applied on the estimated emissions to derive the interquartile range of emissions. Hence, we can consider that the estimated biomass carbon emissions have a RE of $+1.3_{-2.0}^{+7.1}\%$, and the estimated SOC emissions have a RE of $+0.1_{-0.4}^{+0.6}\%$. Re-arranging the Equation (S8) and applying the two quartile values of RE, we can derive the uncertainty of carbon emissions as $31.0_{28.9}^{31.6}$ Tg C yr⁻¹ (our estimated carbon loss and interquartile range of estimated carbon loss) for biomass carbon loss, $43.9_{43.6}^{44.1}$ Tg C yr⁻¹ for SOC loss of the upper boundary, and $14.8_{14.7}^{14.9}$ Tg C yr⁻¹ for SOC loss of the lower boundary, respectively. Because the distribution of RE is represented by its interquartile range, these intervals for biomass and SOC losses suggest that there is a 50% probability that the true value of emissions will fall within these intervals. Furthermore, the uncertainty for global ISA-driven carbon emission can be calculated by simply adding the intervals of biomass with those of upper and lower estimates of SOC emissions, respectively. Therefore, the global total ISA-driven carbon emissions are $74.9_{72.5}^{75.7}$ Tg C yr⁻¹ and $45.8_{43.6}^{46.5}$ Tg C yr⁻¹ for the upper and the lower boundary estimations, respectively.

Figure R9 Spatial distribution of RE (relative error, in %) for SOC density interpolation on a 5-km grid.

Figure R10 The RE (relative error) of SOC interpolation at the global scale and for different land cover types (a) and its histogram (b). The dots and the error bars in panel a indicate the median value and the interquartile range, respectively.

In response to the referee’s request, we have added the uncertainty evaluation regarding the interpolation of the carbon densities in Supplementary Information 2 and briefly mentioned in the Methods of the revised main text (lines 526–536; lines 559-565).

[Comment 5] The application of the Kaya-like method for attributing changes in ISA-driven terrestrial carbon emissions to socio-economic driving factors does appear somewhat arbitrary. In Equation (3), the denominators can indeed be rearranged to form different driving factors, leading to varying conclusions. For instance, if we rearrange the equation as $E = P * P_u/U * U/\Delta U * \Delta U/P_u * E/P$, the driving factors could be interpreted as urban land per urban population, carbon emission per capita, and so on. Furthermore, it is feasible to incorporate other socio-economic variables into the equation, such as proportions of different economic sectors, electricity consumption, etc. This flexibility in the equation’s formulation could potentially lead to a wide range of interpretations and conclusions. Therefore, it is crucial for the authors to justify their choice of variables and their placement in the equation to ensure the robustness and validity of their findings.

[Response] The Kaya-like decomposition method is commonly applied in literature to attribute the dynamics of urban-related and energy-related carbon emissions (Cai et al., 2018; Jiang and O’Neill, 2017; Zhang et al., 2021). We agree with the referee that indeed there are no universally agreed terms to be included in the decomposition equation, and that some ‘expert judgment’ based on the specific research objective is often needed. Given that, to our knowledge, this is the first study to address global-scale terrestrial carbon emissions from ISA expansion and their underlying socioeconomic drivers, there is perhaps no existing decomposition framework that could be directly used. Nonetheless, the selection of explanatory variables in this study is based on empirical knowledge of the dynamics of urban expansion and urban-related carbon emissions as revealed by previous studies (detailed below).

We attributed changes in ISA-driven terrestrial carbon emissions to five factors as in Equation (3): $E = P \cdot P_u/P \cdot ISA/P_u \cdot \Delta ISA/ISA \cdot E/\Delta ISA$. They were interpreted as total population (P),

urbanization rate ($u=Pu/P$), residential ISA intensity ($r=ISA/Pu$), ISA expansion speed-up factor ($s=\Delta ISA/ISA$), and carbon emission intensity ($e=E/\Delta ISA$). The total population (P) is a component of the original Kaya identity method (Kaya, 1989) and is a fundamental driver for human socioeconomic development. The urbanization rate ($u=Pu/P$) reflects the population structure underlying the expansion of urban areas (Zhang et al., 2021), which likely dominate ISA in this study, although our ISA expansion includes those in both urban and rural areas. Both population (P) and the urbanization rate (u) have been used as key variables to construct the narratives of the Shared Socioeconomic Pathways (SSPs) which are further used to project future anthropogenic CO₂ emissions in the most recent IPCC assessment report (AR6) (Jiang and O'Neill, 2017; Riahi et al., 2017), highlighting their fundamental roles in driving urban expansion and the associated carbon emissions. The residential ISA intensity ($r=ISA/Pu$) represents the ISA per urban capita and measures the 'affluence' level in terms of inhabiting space. Similar factors of urban area per capita have also been used to decompose historical changes in urban energy CO₂ emissions through using a Kaya-like method (Cai et al., 2018), and to project future energy use and the resulted CO₂ emissions for the world's cities (Singh and Kennedy, 2015). As ISA-driven carbon emissions result from ISA expansion (ΔISA) rather than the existing ISA, we use carbon emission intensity ($e=E/\Delta ISA$ rather than E/ISA) to measure emissions per area of expanded ISA. This indicator has policy implications for land use decisions on ISA construction. For instance, to reduce ISA-driven emissions, new ISA should be preferably allocated to land with a low carbon density of both biomass and SOC such as marginal land and abandoned land. Similar indicators of carbon intensity per amount of energy use have been commonly used in Kaya-like frameworks to attribute energy CO₂ emissions dynamics, for example, the carbon intensity of energy consumed (Raupach et al., 2007) and of material weight (Goswein et al., 2019). The ISA expansion speed-up factor ($s=\Delta ISA/ISA$) is indeed a unique factor used in this study to characterize the temporal dynamics of ISA expansion.

We did not use proportions of different economic sectors, or electricity consumption as suggested by the referee. Indeed, these factors are often used in the decomposition analysis for energy CO₂ emissions dynamics. Because different economic sectors have diverse emission intensities per economic output (e.g., high-pollution heavy industries versus almost pollution-free R&D industries), and because different energy forms also have different emission intensities for the same amount of energy use (e.g., coal used as a primary energy source versus electricity generated from renewable sources), the economic structure and energy structure can have important consequences for energy CO₂ emissions. However, their linkages to ISA expansion and the associated terrestrial emissions are much less direct. In addition, rearranging the denominators and numerators as suggested by the referee (we understand that it's just an example) could lead to ambiguous results. For example, E/P can be interpreted as ISA-driven emissions per capita but, unlike in the case of energy CO₂ emissions where all existing population must consume some energy and hence result in certain amount of CO₂ emissions, existing population does not have a direct linkage to ISA expansion.

One could argue that a possibly obvious omission from our decomposition framework is a related term of GDP per capita (GDP/P). However, we note that GDP per capita is highly positively correlated with urbanization rate ($u=Pu/P$) (Chen et al., 2014, <https://ourworldindata.org/grapher/urbanization-vs-gdp>), and we analyzed the temporal dynamics of each driving factor as a function of per capita GDP (Figure 5 in the main text). We then argue

that GDP per capita is implicitly accounted for in our socioeconomic analysis of ISA-driven emissions dynamics.

Finally, using our decomposition framework and the selected explanatory variables, we identified a predictable, logically coherent pattern of ISA-driven emissions as a function of economic development. We acknowledge that this fact is, of course, not a direct proof for the rigidity of our framework, nonetheless a logical consistency seems to have been reached.

In response to the referee's comment, the considerations underlying the choice of our framework described above, is now explained more clearly in the revised manuscript (lines 704–719).

[Comment 6] The authors have stated that the percentages of source land covers contributing to ISA expansion were derived from the ESA CCI product (Line 425-427). However, the ESA CCI product only covers the period from 1992 to 2020. This raises a valid question: how were the percentages of source land covers obtained for the years prior to 1992? This is a crucial detail that needs to be addressed to ensure the comprehensiveness and accuracy of the study. The authors should provide a clear explanation of their methodology for obtaining this data.

[Response] We apologize for the unclear description. In the original manuscript, the static land cover map of 1992 was used to identify the source land covers of ISA expansions for all years prior to 1992 (1986–1992). This essentially assumes that no land cover change occurred during 1986–1992 except for ISA expansion. Although a static land cover map has been used in previous studies to investigate source land covers for future urban expansion (for instance, Chen et al. (2020) applied the ESA CCI land cover map of 2015 to estimate the source land covers of future urban expansion over 2020–2100), we realize that studies of future situations can accommodate a much larger degree of uncertainty than those quantifying past changes. Given that the source land cover information plays a pivotal role in determining the carbon densities of pixels transitioning to ISA (see our detailed response to Comment 4), assuming a constant land cover distribution before 1992 might introduce uncertainties that cannot be quantified.

We have not been able to find any other credible land cover datasets capable of replacing the ESA CCI for the period before 1992 and so, to address the referee's concern regarding the accuracy of the study, we deem that the best option is to remove 1986–1992 from our study period. To further enhance the data reliability and robustness of this study, we ensure that each year of the study period is covered by at least three of the four ISA products used in this study. For 2019, only two ISA products (GISA and ESA CCI) are available while for 2020 only ESA CCI is available. Hence these two years have also been removed from our study domain. A shorter study period (i.e., 1993–2018) than that in our original manuscript does not change the conclusions but has helped to enhance data reliability and the robustness of our findings.

The selection of the study period (1993–2018) is now clearly explained in the revised manuscript (lines 457–461).

[Comment 7] The acceleration rate of ISA expansion in this study, stated as $180.2 \text{ km}^2 \text{ yr}^{-2}$ (Line 110 and Figure 1-a), does seem puzzling. Both the studies releasing the GAIA and GAUD datasets report a steady growth of global ISA with no noticeable acceleration over the 30-year period. This discrepancy warrants further clarification.

[Response] Following the referee's comment, we have double-checked the calculated ISA expansions using the four ISA products (i.e., GAUD, GAIA, GLC_FCS30, and ESA CCI) and found that the derived areas of ISA expansion in the original manuscript were incorrect. This issue arose because the nominal resolution based on the geographic coordinates (in degrees), rather than the projected coordinates (in meters), was used to calculate the area of ISA. This has led to an overestimation of ISA extent, particularly in the high latitudes. Correcting this error led to a revised ISA expansion acceleration rate of $135.0 \text{ km}^2 \text{ yr}^{-2}$ ($p < 0.05$) for 1986–2020 using the four products as in the original manuscript, instead of the originally reported $180.2 \text{ km}^2 \text{ yr}^{-2}$ ($p < 0.05$). Note that the annual ISA areas derived from GAUD differ slightly from those reported in the original study because we used a newer version of this product than the original study (Liu et al., 2020).

Although the acceleration of global ISA expansion becomes smaller after error correction, it remains at odds with the absence of such a noticeable acceleration reported in GAIA and GAUD as mentioned by the referee. A closer examination shows that the remarkable acceleration derived using the average of the four ISA products during 1986–2020 was primarily attributed to GLC_FCS30, which shows an acceleration of $380.5 \text{ km}^2 \text{ yr}^{-2}$ ($p < 0.01$) in global ISA expansion, while the other ISA products did not show significant acceleration. This indicates that the temporal trend in ISA expansion derived by GLC_FCS30 may be an outlier. This is possibly because the GLC_FCS30 product is originally provided with a 5-year time step, which was disaggregated into annual time step based on the average annual ISA changes derived from the other three annual products. Given that the actual annual ISA expansions remains unknown according to GLC_FCS30 and the temporal disaggregation introduces uncertainty, we decided to remove GLC_FCS30 from our analysis, and instead, added a new annual ISA dataset (GISA2.0) as pointed out by Referee #1. Indeed, replacing GLC_FCS30 with GISA results in an insignificant temporal trend of $31.1 \text{ km}^2 \text{ yr}^{-2}$ ($p > 0.05$) in global ISA expansion during 1986–2020 using the average of the four ISA products (GAUD, GAIA, GISA2.0, and ESA CCI).

In the revised manuscript, we have changed the study period to 1993–2018 for two considerations (see our detailed response to Comment 6 above): 1) annual land cover maps prior to 1992 are not available; and 2) each year of the study period is covered by at least three ISA products to reduce the uncertainty in the derived ISA expansion. Our analysis with the new study period shows a significant acceleration rate of $305.5 \text{ km}^2 \text{ yr}^{-2}$ ($p < 0.01$) in global ISA expansion using the average of the four ISA products (i.e., GAUD, GAIA, ESA CCI and GISA2.0). This acceleration rate was attributed to a significant temporal trend in global ISA expansion according to GAIA ($664.7 \text{ km}^2 \text{ yr}^{-2}$, $p < 0.01$) and GISA ($416.7 \text{ km}^2 \text{ yr}^{-2}$, $p < 0.01$), an insignificant trend according to ESA CCI ($221.1 \text{ km}^2 \text{ yr}^{-2}$, $p = 0.23$), all being partially offset by a negative but insignificant temporal trend according to GAUD ($-220.7 \text{ km}^2 \text{ yr}^{-2}$, $p = 0.10$). For 1986–2018 over which at least three ISA products are available, the temporal trend in global ISA expansion is $78.2 \text{ km}^2 \text{ yr}^{-2}$ ($p > 0.05$), being almost the double of that over 1986–2020 but remaining insignificant.

These results imply that, depending on the time period being used, the interpretations on whether global ISA expansion shows acceleration can be different: the temporal trends in global ISA

expansion is $305.5 \text{ km}^2 \text{ yr}^{-2}$ for 1993–2018 ($p < 0.01$) and $78.2 \text{ km}^2 \text{ yr}^{-2}$ ($p > 0.05$) for 1986–2018, with both periods being covered by at least three ISA products. Although the acceleration persists for both periods, but it is not significant over the longer period of 1986–2018.

We acknowledge that for trend analysis using time-series data, it is common that the selected time window can affect the outcomes. Nonetheless, we argue that the acceleration of global ISA expansion over 1993–2018 is not really the key finding and the novelty of this study. Instead, our key novelty is that this study provides the first estimate of terrestrial carbon losses due to global ISA expansion. Another relevant key finding is that the ISA-driven carbon emissions dynamics depend on economic development stages of different countries.

To evaluate whether the selection of the study period affects the key findings of this study mentioned above, we calculated ISA-driven carbon emissions for 1986–2018 using the method as described in the revised manuscript, but assuming constant land cover distribution of 1992 ESA CCI land cover for the period of 1986–1992. As carbon densities for pixels transitioning to ISA highly depend on their source land covers, we acknowledge that the derived emissions for 1986–1992 might be not accurate. With this uncertainty in mind, we performed the same analysis to generate the equivalent figures for Fig. 2, Fig. 4, and Fig. 5 in the main text but using the results for 1986–2018 (Figures R11, R12 and R13).

The results show that the ISA expansion and the associated carbon emission over 1986–2018 show similar patterns as over 1993–2018, for leading countries and country groups as well as in terms of their spatial distributions (compare Figure R11 and main text Fig. 2). More importantly, the contrasting temporal trends in ISA-driven emissions between NAI and AI countries, and the relative importance of their underlying drivers remain almost the same for 1986–2018 as for 1993–2018 (Figure R12). It is comprehensible that the relative growth rates of ISA expansion speed-up factor (s) are different in Figure R12 and main text Fig. 4 because ISA expansion over the two periods are different. Moreover, the relationships between the relative change rates in the carbon emissions, as well as their underlying driving variables (P , u , r , s , e), and $\log_{10}(\text{GDP}_{\text{cap}})$ derived using the results for 1986–2018 (Figure R13) highly resemble those derived using the results for 1993–2018 (main text Fig. 5), confirming the robustness of the finding of a predictable emissions dynamics pattern depending on economic development stage.

Based on these results, we conclude that although using the study period of 1993–2018 leads to a different temporal trend of global ISA expansion than using the whole period of 1986–2018, it has no influence on our key findings regarding emissions patterns for different nations, emissions trends between AI and NAI countries and the identified predictable pattern of emissions dynamics as a function of per capita GDP. To ensure the accuracy in the calculation of emissions, we decide to use the period of 1993–2018 in the revised main text (see our response to Comment 6).

Figure R11 Impervious surface area expansion and the associated carbon emissions during 1986–2018 for leading countries and country groups, and their spatial distributions (the same figure as main text Fig. 2 but using the results of 1986–2018).

Figure R12 Drivers of the dynamics in the carbon emissions due to ISA expansion over 1986–2018 (the same figure as main text Fig. 4 but using the results over 1986–2018).

Figure R13 Relationships between the relative change rates in ISA-driven carbon emissions and the underlying drivers over 1986–2018 and per capita GDP (the same figure as main text Fig. 5 but using the results over 1986–2018).

[Comment 8] The availability of live biomass carbon density maps only for the year 2010, and the lack of a time-series dataset for SOC density (SoilGrids250m), raises questions about how the authors derived the carbon density of transition lands during the study period from 1986 to 2020. This aspect needs to be addressed for a comprehensive understanding of the results and methodology.

[Response] As is suggested by the referee, the uncertainty in the quantified ISA-driven terrestrial carbon losses, contributed by the employed biomass and SOC density maps, has two main sources: (1) static rather than dynamic carbon density maps are used; (2) spatial interpolation has been used to derive carbon densities for the pixels transitioning to ISA.

(1) Dynamic carbon density maps

Following the referee's suggestion, we tested using dynamic (annual) woody (forest and shrubland), cropland and grassland biomass density maps to quantify biomass carbon losses driven by ISA expansion. Using the best available remote sensing data, we found that using dynamic biomass density maps brings only negligible changes in the derived biomass carbon losses compared to using static maps. In the case of woody biomass losses, the difference in the derived biomass carbon losses obtained by using different static maps far exceeds that obtained by using dynamic versus static biomass density maps. Hence, we finally decided to continue using static biomass density maps in our revised manuscript (but for woody biomass, three best available static maps are used to reduce the uncertainty). Please also refer to our responses to Comment 2 and Comment 3 above.

We did not make any tests using dynamic SOC density maps, because in contrast to the case for biomass, we are not aware of any observation-based, highly credible and widely used dynamic SOC density maps. Some annual SOC density maps have been reported in the literature: for example, Sanderman et al. (2017) generated 10-km global SOC density maps for the past 12,000 years until 2010, and Zhao et al. (2021) (<https://zenodo.org/records/5040380>) provide 5-km global SOC density maps for 1981–2018. However, neither of these datasets is suitable for our application. Both datasets are generated using a space-for-time (SFT) substitution modeling approach, i.e., the relationships between site-level static SOC measurements and multiple covariate climate and environmental variables (topography, vegetation, land cover, soil property etc.) are used to extrapolate dynamic SOC densities over time through applying dynamic covariate variables (climate, land cover, etc.) in the process of prediction. But warnings are frequently given regarding the usage of the SFT substitution to predict temporal changes because it can lead to spurious changes that are highly impacted by static spatial gradients, rather than actual changes over time (Yue et al., 2024). Although Zhao et al. (2021) additionally used process-based SOC models in their dynamic SOC prediction to alleviate the disadvantages of the SFT approach, this does not fundamentally change the nature of their prediction.

In addition, the dataset of Sanderman et al. (2017) focused on quantifying historical SOC changes caused by human land use and land cover change, whereas in our application we are more concerned with dynamic SOC densities given the same land cover, because, in our spatial interpolation of SOC density, only SOC values of the same land cover have been used to interpolate for the pixel transitioning to ISA. Given the same land cover, the SFT substitution approach used in Sanderman et al. and Zhao et al. is more likely to yield spurious long-term trends in SOC, mostly driven by anthropogenic climate change when more local-scale impacts by other

factors (management changes, selective logging of forest, etc.) are omitted in their models. This will subsequently introduce artificial trends in the SOC losses caused by ISA expansion. Moreover, to our knowledge, neither Sanderman et al. (2017) nor Zhao et al. (2021) underwent rigorous validation in terms of their dynamic SOC changes over time. Lastly, given the same land cover type, changes in SOC are expected to lag behind changes in live biomass, because there is a delay in the transfer of altered biomass litter input into soil and its subsequent stabilization in the form of soil organic matter. As we see little impact on the derived biomass losses through using remote sensing-based dynamic biomass maps, we expect that the impact on the derived SOC losses, if dynamic SOC density maps congruent with remote sensing biomass changes were available, would probably have been even smaller.

Based on the above considerations, we expect that although constructing dynamic SOC density maps, through combining the SoilGrids250m product with the above-mentioned coarse-resolution (5 km to 10 km) dynamic SOC density maps, is practically doable, it will be unlikely to bring useful insights on reducing the associated uncertainty. Hence, we have retained the static SoilGrids250m product, the only widely used, highly authoritative, medium-resolution SOC product covering the globe, in our revised manuscript.

(2) Spatial interpolation of carbon density for pixels transitioning to ISA

Following the referee's suggestion, we have now quantified the uncertainty associated with the spatial interpolation of carbon density for pixels transitioning to ISA. Please refer to our response to Comment 4 above.

[Comment 9] The choice to treat the loss ratio of the IPCC default value as the lower boundary and the value derived from literature synthesis as the upper boundary for SOC loss is not immediately clear. The underlying logic behind this setup should be explained to ensure transparency in the methodology.

[Response] The IPCC guidelines suggest that for land converted to settlements that are paved over (i.e., impervious surface areas), 20% of the soil carbon relative to the previous land use is considered as being lost as a result of disturbance, removal or relocation. However, to our knowledge, this suggested loss ratio lacks justification or support from site-level observations.

In contrast, our synthesis of global site-level measurements of SOC loss following ISA establishment shows a mean SOC loss ratio of 59.5% (Supplementary Fig. 12), being much higher than the IPCC default value. In fact, of the 22 collected measurements, only one shows a SOC loss ratio (17.2%) close to the IPCC default value, with all other values being higher than the IPCC default one (the second lowest loss ratio is 33.7%). We also notice that almost all the studies in our synthesis were published within the last four years. Given that the sources of the IPCC default value remain unknown, there is a good chance that the actual SOC loss ratio is much larger than the IPCC default value. In addition, a nationwide study from Ding et al. (2022) of China incorporating 148 paired measurements of SOC for soil depths of up to 100 cm on adjacent sites of ISA and pervious areas (i.e., lawns, bare land, etc.) reported that, on average, SOC under ISA was 38% lower than those in pervious areas (unfortunately their data have not been included in our dataset because they are not available). This result further supports the suggestion that the default IPCC value is probably too low. The SOC loss ratios of 38% found by Ding et al. (2022) for China and of 59.5% across the northern hemisphere by our study are highly credible, as they

are close to the SOC loss ratios derived from permanent bare fallow experiments (Supplementary Fig. 12), which share a similar mechanism underlying SOC loss to that due to ISA expansion.

But, on the other hand, both our synthesis of studies and the study of China by Ding et al. (2022) have a focus on the northern hemisphere. Observations of SOC loss driven by ISA expansion from warm and wet tropical regions are particularly needed in the future. Using the IPCC default value of 20% will almost certainly underestimate SOC loss following ISA expansion, whereas using the mean value of 59.5% from our synthesis will have a strong northern hemisphere bias. We therefore adopted a somewhat compromised approach to use the IPCC default value as the lower bound, to be comparable with that used in national greenhouse gas inventories, and to use our derived value as the upper bound, to account for actual observations.

The elaborations made above have been incorporated in the revised manuscript (lines 630–647), to clarify the considerations behind our choice of using upper and lower bounds to estimate ISA-driven SOC losses.

Appendix Table 1 Literature review focusing on the topic of biomass loss due to land cover change using satellite datasets.

	Reference	Research objective	Biomass density used	Method
1	Harris et al. (2021)	Estimate annual forest-related greenhouse gas emissions and removals induced by forest change globally over 2001–2019.	Use a static forest aboveground live biomass density map in the base year of 2000 with a resolution of 30 m.	1
2	Feng et al. (2022)	Estimate annual forest carbon losses associated with forest removal over the tropics over 2001–2020.	Use four static AGB density maps with different resolutions (1-km, 1-km, 500-m, and 30-m) mostly across the tropics.	1
3	Feng et al. (2021)	Estimate annual forest carbon losses due to forest loss over the mountains of Southeast Asia from 2001–2019.	Use the static forest carbon stock density map circa 2000 with the resolution of 30 m.	1
4	Lai et al. (2016)	Estimate terrestrial carbon stock changes due to land-use change in China during 1990–2010.	Use the biomass/SOC carbon density values for each difference vegetation/soil type from literature review.	1

5	Xu et al. (2022)	Estimate carbon losses due to deforestation from oil palm expansion during 2001–2015 in Indonesia and Malaysia.	Use a static biomass density map (GlobBiomass2010) with a resolution of 100 m in 2010. The forest AGB density for pixels with oil palm expansion was interpolated using nearest-neighbor method.	1
6	Richards et al. (2020)	Estimate net changes of the mangrove carbon stock globally due to land cover change between 1996 and 2016.	Static above- and belowground tree biomass carbon density maps were averaged at the 0.05- or 0.5-degree grid cells overlaid with mangrove loss or gain data in the same grid cell to estimate the mangrove carbon stock changes. The paper reported in the Methods part that carbon sequestration was not considered because of the unavailability of global high-resolution maps of mangrove carbon sequestration rates.	1
7	Duncanson et al. (2023)	Estimate the carbon stock changes in protected areas globally between 2000 and 2020.	Use a static biomass density map at the resolution of 1 km from GEDI, which utilized its mission data between April 2019 and September 2020.	1
8	Estoque et al. (2019)	Estimate the aboveground forest carbon stock changes due to forest change from 2005 to 2015 over the forests in Southeast Asia.	Use a static AGB density map for 2000 with the resolution of 30 m.	1
9	Zhu et al. (2021)	Estimate the carbon stock changes of the ecosystem post-2020 across the Asia region.	Use the static AGB and BGB carbon density maps from Spawn et al. (2020) in 2010 at 300-m resolution.	1
10	Uribe et al. (2023)	Estimate the net loss of biomass from the neotropics due to the contraction of humid regions and expansion of those with intense dry periods over 1950–2100.	The average AGB within each climatic zone (totally 7 zones) from 3 static AGB maps (with the resolutions of 500 m, 10 km, and 100 m) circa 2010 was directly assigned to the future pixels according to the relationship between AGB and climate (aridity index) built in the historical period.	1

11	Maxwell et al. (2019)	Estimate the biomass carbon loss due to degradation and forgone removals of intact forest loss across the pantropics during 2000–2013.	Three static pantropical aboveground biomass maps of woody vegetation were used, one with 1-km resolution for the 2000s, and the other two with 1-km and 463-m resolutions, respectively.	1
12	Fan et al. (2023)	Analyze annual live and dead aboveground carbon changes from 2010 to 2019 in Siberian forests.	3 static AGB carbon density maps (for 2010, 2015, and 2017, respectively) were applied to calibrate an empirical function of yearly L-VOD and aboveground carbon density to derive annual AGB carbon density maps at 25-km resolution.	2
13	Zhang et al. (2022)	Estimate the terrestrial carbon budget due to urbanization and rural depopulation in China during 2002–2019.	Apply annual aboveground carbon density maps by using a static benchmark map of aboveground carbon density map (1-km resolution, 2018) and random forest modeling with annual MODIS data.	2
14	Yang et al. (2023)	Estimate terrestrial carbon storage change under environmental and land-use changes over 2010–2019 globally.	Use global maps of annual live vegetation biomass at 25-km resolution using L-band microwave vegetation optical depth. The spatial calibration functions relating L-VOD to AGB with four existing tropical/global AGB maps at 25-km resolution (the original resolution of L-VOD) were built to derive annual biomass maps.	2

Note: Method 1 indicates using static biomass carbon density maps overlying land cover change maps; Me 2 indicates biomass mapping without using land cover maps.

References

- Cai, B., Li, W., Dhakal, S., Wang, J., 2018. Source data supported high resolution carbon emissions inventory for urban areas of the Beijing-Tianjin-Hebei region: Spatial patterns, decomposition and policy implications. *Journal of Environmental Management* 206, 786–799. <https://doi.org/10.1016/j.jenvman.2017.11.038>
- Ch, R., Martin, D.A., Vargas, J.F., 2021. Measuring the size and growth of cities using nighttime light. *Journal of Urban Economics, Delineation of Urban Areas* 125, 103254. <https://doi.org/10.1016/j.jue.2020.103254>
- Chapin, F.S., Woodwell, G.M., Randerson, J.T., Rastetter, E.B., Lovett, G.M., Baldocchi, D.D., Clark, D.A., Harmon, M.E., Schimel, D.S., Valentini, R., Wirth, C., Aber, J.D., Cole, J.J., Goulden, M.L., Harden, J.W., Heimann, M., Howarth, R.W., Matson, P.A., McGuire, A.D., Melillo, J.M., Mooney, H.A., Neff, J.C., Houghton, R.A., Pace, M.L., Ryan, M.G., Running, S.W., Sala, O.E., Schlesinger, W.H., Schulze, E.-D., 2006. Reconciling Carbon-cycle Concepts, Terminology, and Methods. *Ecosystems* 9, 1041–1050. <https://doi.org/10.1007/s10021-005-0105-7>
- Chen, M., Zhang, H., Liu, W., Zhang, W., 2014. The global pattern of urbanization and economic growth: evidence from the last three decades. *PloS one* 9, e103799.
- Cummins, S., Fagg, J., 2012. Does greener mean thinner? Associations between neighbourhood greenspace and weight status among adults in England. *Int J Obes* 36, 1108–1113. <https://doi.org/10.1038/ijo.2011.195>
- Dallimer, M., Tang, Z., Bibby, P.R., Brindley, P., Gaston, K.J., Davies, Z.G., 2011. Temporal changes in greenspace in a highly urbanized region. *Biology Letters* 7, 763–766. <https://doi.org/10.1098/rsbl.2011.0025>
- de Bellefon, M.-P., Combes, P.-P., Duranton, G., Gobillon, L., Gorin, C., 2021. Delineating urban areas using building density. *Journal of Urban Economics, Delineation of Urban Areas* 125, 103226. <https://doi.org/10.1016/j.jue.2019.103226>
- Ding, Q., Shao, H., Chen, X., Zhang, C., 2022. Urban Land Conversion Reduces Soil Organic Carbon Density Under Impervious Surfaces. *Global Biogeochemical Cycles* 36, e2021GB007293. <https://doi.org/10.1029/2021GB007293>
- Duncanson, L., Liang, M., Leitold, V., Armston, J., Krishna Moorthy, S.M., Dubayah, R., Costedoat, S., Enquist, B.J., Fatoyinbo, L., Goetz, S.J., 2023. The effectiveness of global protected areas for climate change mitigation. *Nature Communications* 14, 2908.
- Edmondson, J.L., Davies, Z.G., McCormack, S.A., Gaston, K.J., Leake, J.R., 2014. Land-cover effects on soil organic carbon stocks in a European city. *Science of The Total Environment* 472, 444–453. <https://doi.org/10.1016/j.scitotenv.2013.11.025>
- Estoque, R.C., Ooba, M., Avitabile, V., Hijioka, Y., DasGupta, R., Togawa, T., Murayama, Y., 2019. The future of Southeast Asia's forests. *Nature communications* 10, 1829.
- Fan, L., Wigneron, J.-P., Ciais, P., Chave, J., Brandt, M., Sitch, S., Yue, C., Bastos, A., Li, X., Qin, Y., 2023. Siberian carbon sink reduced by forest disturbances. *Nature Geoscience* 16, 56–62.
- Feng, Y., Zeng, Z., Searchinger, T.D., Ziegler, A.D., Wu, J., Wang, D., He, X., Elsen, P.R., Ciais, P., Xu, R., 2022. Doubling of annual forest carbon loss over the tropics during the early twenty-first century. *Nature Sustainability* 5, 444–451.
- Feng, Y., Ziegler, A.D., Elsen, P.R., Liu, Y., He, X., Spracklen, D.V., Holden, J., Jiang, X., Zheng, C., Zeng, Z., 2021. Upward expansion and acceleration of forest clearance in the mountains of Southeast Asia. *Nature Sustainability* 4, 892–899.
- Goswein, V., Silvestre, J.D., Habert, G., Freire, F., 2019. Dynamic assessment of construction materials in urban building stocks: a critical review. *Environmental science & technology* 53, 9992–10006.
- Guo, H., Du, E., Terrer, C., Jackson, R.B., 2024. Global distribution of surface soil organic carbon in urban greenspaces. *Nat Commun* 15, 806. <https://doi.org/10.1038/s41467-024-44887-y>
- Harris, N.L., Gibbs, D.A., Baccini, A., Birdsey, R.A., de Bruin, S., Farina, M., Fatoyinbo, L., Hansen, M.C., Herold, M., Houghton, R.A., Potapov, P.V., Suarez, D.R., Roman-Cuesta, R.M., Saatchi, S.S., Slay, C.M., Turubanova, S.A., Tyukavina, A., 2021. Global maps of twenty-first century

- forest carbon fluxes. *Nat. Clim. Chang.* 11, 234–240. <https://doi.org/10.1038/s41558-020-00976-6>
- Jiang, L., O’Neill, B.C., 2017. Global urbanization projections for the Shared Socioeconomic Pathways. *Global Environmental Change* 42, 193–199.
- Kabisch, N., Haase, D., 2013. Green spaces of European cities revisited for 1990–2006. *Landscape and Urban Planning* 110, 113–122. <https://doi.org/10.1016/j.landurbplan.2012.10.017>
- Kaya, Y., 1989. Impact of carbon dioxide emission control on GNP growth: interpretation of proposed scenarios. Intergovernmental Panel on Climate Change/Response Strategies Working Group, May.
- Lai, L., Huang, X., Yang, H., Chuai, X., Zhang, M., Zhong, T., Chen, Z., Chen, Y., Wang, X., Thompson, J.R., 2016. Carbon emissions from land-use change and management in China between 1990 and 2010. *Science Advances* 2, e1601063. <https://doi.org/10.1126/sciadv.1601063>
- Li, Xuecao, Gong, P., Zhou, Y., Wang, J., Bai, Y., Chen, B., Hu, T., Xiao, Y., Xu, B., Yang, J., Liu, X., Cai, W., Huang, H., Wu, T., Wang, X., Lin, P., Li, Xun, Chen, J., He, C., Li, Xia, Yu, L., Clinton, N., Zhu, Z., 2020. Mapping global urban boundaries from the global artificial impervious area (GAIA) data. *Environ. Res. Lett.* 15, 094044. <https://doi.org/10.1088/1748-9326/ab9be3>
- Liu, X., Huang, Y., Xu, X., Li, Xuecao, Li, Xia, Ciais, P., Lin, P., Gong, K., Ziegler, A.D., Chen, A., 2020. High-spatiotemporal-resolution mapping of global urban change from 1985 to 2015. *Nature Sustainability* 3, 564–570.
- Liu, X., Pei, F., Wen, Y., Li, X., Wang, S., Wu, C., Cai, Y., Wu, J., Chen, J., Feng, K., Liu, J., Hubacek, K., Davis, S.J., Yuan, W., Yu, L., Liu, Z., 2019. Global urban expansion offsets climate-driven increases in terrestrial net primary productivity. *Nat Commun* 10, 5558. <https://doi.org/10.1038/s41467-019-13462-1>
- Liu, Z., He, C., Zhou, Y., Wu, J., 2014. How much of the world’s land has been urbanized, really? A hierarchical framework for avoiding confusion. *Landscape Ecol* 29, 763–771. <https://doi.org/10.1007/s10980-014-0034-y>
- Maxwell, S.L., Evans, T., Watson, J.E., Morel, A., Grantham, H., Duncan, A., Harris, N., Potapov, P., Runting, R.K., Venter, O., 2019. Degradation and forgone removals increase the carbon impact of intact forest loss by 626%. *Science Advances* 5, eaax2546.
- Pouyat, R.V., Yesilonis, I.D., Golubiewski, N.E., 2009. A comparison of soil organic carbon stocks between residential turf grass and native soil. *Urban Ecosyst* 12, 45–62. <https://doi.org/10.1007/s11252-008-0059-6>
- Raupach, M.R., Marland, G., Ciais, P., Le Quéré, C., Canadell, J.G., Klepper, G., Field, C.B., 2007. Global and regional drivers of accelerating CO₂ emissions. *Proceedings of the National Academy of Sciences* 104, 10288–10293.
- Riahi, K., van Vuuren, D.P., Kriegler, E., Edmonds, J., O’Neill, B.C., Fujimori, S., Bauer, N., Calvin, K., Dellink, R., Fricko, O., Lutz, W., Popp, A., Cuaresma, J.C., Kc, S., Leimbach, M., Jiang, L., Kram, T., Rao, S., Emmerling, J., Ebi, K., Hasegawa, T., Havlik, P., Humpenöder, F., Da Silva, L.A., Smith, S., Stehfest, E., Bosetti, V., Eom, J., Gernaat, D., Masui, T., Rogelj, J., Strefler, J., Drouet, L., Krey, V., Luderer, G., Harmsen, M., Takahashi, K., Baumstark, L., Doelman, J.C., Kainuma, M., Klimont, Z., Marangoni, G., Lotze-Campen, H., Obersteiner, M., Tabeau, A., Tavoni, M., 2017. The Shared Socioeconomic Pathways and their energy, land use, and greenhouse gas emissions implications: An overview. *Global Environmental Change* 42, 153–168. <https://doi.org/10.1016/j.gloenvcha.2016.05.009>
- Richards, D.R., Thompson, B.S., Wijedasa, L., 2020. Quantifying net loss of global mangrove carbon stocks from 20 years of land cover change. *Nat Commun* 11, 4260. <https://doi.org/10.1038/s41467-020-18118-z>
- Sanderman, J., Hengl, T., Fiske, G.J., 2017. Soil carbon debt of 12,000 years of human land use. *Proceedings of the National Academy of Sciences* 114, 9575–9580. <https://doi.org/10.1073/pnas.1706103114>

- Santoro, M., Cartus, O., 2023. ESA Biomass Climate Change Initiative (Biomass_cci): Global datasets of forest above-ground biomass for the years 2010, 2017, 2018, 2019 and 2020, v4. <https://doi.org/10.5285/AF60720C1E404A9E9D2C145D2B2EAD4E>
- Seto, K.C., Güneralp, B., Hutyrá, L.R., 2012. Global forecasts of urban expansion to 2030 and direct impacts on biodiversity and carbon pools. *Proc. Natl. Acad. Sci. U.S.A.* 109, 16083–16088. <https://doi.org/10.1073/pnas.1211658109>
- Singh, S., Kennedy, C., 2015. Estimating future energy use and CO₂ emissions of the world's cities. *Environmental Pollution* 203, 271–278.
- Spawn, S.A., Sullivan, C.C., Lark, T.J., Gibbs, H.K., 2020. Harmonized global maps of above and belowground biomass carbon density in the year 2010. *Sci Data* 7, 112. <https://doi.org/10.1038/s41597-020-0444-4>
- Sun, L., Chen, J., Li, Q., Huang, D., 2020. Dramatic uneven urbanization of large cities throughout the world in recent decades. *Nature communications* 11, 5366.
- Taylor, L., Hochuli, D.F., 2017. Defining greenspace: Multiple uses across multiple disciplines. *Landscape and Urban Planning* 158, 25–38. <https://doi.org/10.1016/j.landurbplan.2016.09.024>
- Uchiyama, Y., Mori, K., 2017. Methods for specifying spatial boundaries of cities in the world: The impacts of delineation methods on city sustainability indices. *Science of The Total Environment* 592, 345–356. <https://doi.org/10.1016/j.scitotenv.2017.03.014>
- Uribe, M. del R., Coe, M.T., Castanho, A.D., Macedo, M.N., Valle, D., Brando, P.M., 2023. Net loss of biomass predicted for tropical biomes in a changing climate. *Nature Climate Change* 13, 274–281.
- Vasenev, V., Kuzyakov, Y., 2018. Urban soils as hot spots of anthropogenic carbon accumulation: Review of stocks, mechanisms and driving factors. *Land Degrad Dev* 29, 1607–1622. <https://doi.org/10.1002/ldr.2944>
- Wu, L., Kim, S.K., 2021. Exploring the equality of accessing urban green spaces: A comparative study of 341 Chinese cities. *Ecological Indicators* 121, 107080. <https://doi.org/10.1016/j.ecolind.2020.107080>
- Xu, L., Saatchi, S.S., Yang, Y., Yu, Y., Pongratz, J., Bloom, A.A., Bowman, K., Worden, J., Liu, J., Yin, Y., 2021. Changes in global terrestrial live biomass over the 21st century. *Science Advances* 7, eabe9829.
- Xu, Y., Yu, L., Ciais, P., Li, W., Santoro, M., Yang, H., Gong, P., 2022. Recent expansion of oil palm plantations into carbon-rich forests. *Nature Sustainability* 5, 574–577.
- Yang, H., Ciais, P., Frappart, F., Li, X., Brandt, M., Fensholt, R., Fan, L., Saatchi, S., Besnard, S., Deng, Z., Bowring, S., Wigneron, J.-P., 2023. Global increase in biomass carbon stock dominated by growth of northern young forests over past decade. *Nat. Geosci.* 16, 886–892. <https://doi.org/10.1038/s41561-023-01274-4>
- Yue, C., Jian, J., Ciais, P., Ren, X., Jiao, J., An, S., Li, Y., Wu, J., Zhang, P., Bond-Lamberty, B., 2024. Field experiments show no consistent reductions in soil microbial carbon in response to warming. *Nat Commun* 15, 1731. <https://doi.org/10.1038/s41467-024-45508-4>
- Zhang, D., Wang, Z., Li, S., Zhang, H., 2021. Impact of Land Urbanization on Carbon Emissions in Urban Agglomerations of the Middle Reaches of the Yangtze River. *International Journal of Environmental Research and Public Health* 18, 1403. <https://doi.org/10.3390/ijerph18041403>
- Zhang, X., Brandt, M., Tong, X., Ciais, P., Yue, Y., Xiao, X., Zhang, W., Wang, K., Fensholt, R., 2022. A large but transient carbon sink from urbanization and rural depopulation in China. *Nat Sustain* 5, 321–328. <https://doi.org/10.1038/s41893-021-00843-y>
- Zhang, Z., Gao, X., Zhang, S., Gao, H., Huang, J., Sun, S., Song, X., Fry, E., Tian, H., Xia, X., 2022. Urban development enhances soil organic carbon storage through increasing urban vegetation. *Journal of Environmental Management* 312, 114922. <https://doi.org/10.1016/j.jenvman.2022.114922>

- Zhao, Y.C., Xie, E.Z., Zhang, X., Peng, Y.X., 2021. Global topsoil SOC stock from 1981 to 2018 estimated by combining process-based model and space-for-time digital soil mapping. Zenodo, v 1.
- Zhu, L., Hughes, A.C., Zhao, X.-Q., Zhou, L.-J., Ma, K.-P., Shen, X.-L., Li, S., Liu, M.-Z., Xu, W.-B., Watson, J.E., 2021. Regional scalable priorities for national biodiversity and carbon conservation planning in Asia. *Science Advances* 7, eabe4261.
- Zhuang, Q., Shao, Z., Li, D., Huang, X., Li, Y., Altan, O., Wu, S., 2023. Impact of global urban expansion on the terrestrial vegetation carbon sequestration capacity. *Science of The Total Environment* 879, 163074. <https://doi.org/10.1016/j.scitotenv.2023.163074>

REVIEWER COMMENTS

Reviewer #1 (Remarks to the Author):

The authors answered my previous concerns about the article. Therefore, I recommend the manuscript for publication in Nature Communications.

Reviewer #2 (Remarks to the Author):

I have meticulously reviewed the authors' response letter and the revised manuscript. The authors have made substantial revisions to address the uncertainties related to carbon data sources, the selection of ISA data, and the scale effect. The majority of my previous concerns have been satisfactorily addressed.

However, I still have two remaining queries:

(1) In the response letter, the authors identified that the fraction of ISA within a 300m pixel (the spatial resolution of the ESA CCI land cover product) can introduce biases in the estimated biomass carbon loss. They classified pixels with an ISA fraction greater than 5% as ISA-contaminated and employed an IDW method to interpolate the carbon density in these ISA-contaminated pixels from the carbon densities of nearby ISA-free (ISA fraction less than 5%) pixels. If my understanding is correct, the carbon density in the ISA-contaminated pixel should be interpolated from other pre-established (true) carbon densities also in pixels with an ISA fraction greater than 5%, rather than in pixels with an ISA fraction less than 5%. However, there is a lack or even absence of pre-established (true) carbon densities in pixels with an ISA fraction less than 5%. A potential alternative solution to approximate the carbon density in ISA-contaminated pixels might be to assume that the biomass carbon loss is proportional to the presence of ISA fraction within the 300m pixel.

(2) The primary objective of this study is to quantify the CO₂ release from the original organic matter in plant biomass, surface litter, and mineral soil of previous plant ecosystems that are replaced by ISA, a factor previously overlooked in global budget assessments. The authors' use of the term 'carbon loss' in the title and abstract could potentially lead to confusion about whether or not such carbon loss includes NPP or NEP lost due to the replacement of ISA. I would recommend that the authors revise the title and abstract to avoid such potential misunderstandings.

Reviewer #1(Remarks to the Author):

The authors answered my previous concerns about the article. Therefore, I recommend the manuscript for publication in Nature Communications.

[Response] Once again, we would like to express our gratitude to the referee for the constructive feedback, which has helped to substantially improve the quality of the manuscript.

Reviewer #2 (Remarks to the Author):

I have meticulously reviewed the authors' response letter and the revised manuscript. The authors have made substantial revisions to address the uncertainties related to carbon data sources, the selection of ISA data, and the scale effect. The majority of my previous concerns have been satisfactorily addressed.

However, I still have two remaining queries:

[Comment 1] In the response letter, the authors identified that the fraction of ISA within a 300m pixel (the spatial resolution of the ESA CCI land cover product) can introduce biases in the estimated biomass carbon loss. They classified pixels with an ISA fraction greater than 5% as ISA-contaminated and employed an IDW method to interpolate the carbon density in these ISA-contaminated pixels from the carbon densities of nearby ISA-free (ISA fraction less than 5%) pixels. If my understanding is correct, the carbon density in the ISA-contaminated pixel should be interpolated from other pre-established (true) carbon densities also in pixels with an ISA fraction greater than 5%, rather than in pixels with an ISA fraction less than 5%. However, there is a lack or even absence of pre-established (true) carbon densities in pixels with an ISA fraction less than 5%. A potential alternative solution to approximate the carbon density in ISA-contaminated pixels might be to assume that the biomass carbon loss is proportional to the presence of ISA fraction within the 300m pixel.

[Response] We are grateful for confirmation that the concerns raised in the first round of review have been satisfactorily addressed. In this response, we first clarify the interpolation method applied in the manuscript (section 1). Then, we demonstrate that the observed AGB was influenced by sub-pixel ISA presence (section 2.1) and show this can be successfully mitigated by the interpolation method applied in our manuscript (section 2.2). In section 2.3, we explore the alternative approach suggested by the referee, which inspired us to develop a slope correction-based method to mitigate the influence of ISA on observed AGB. We found that the ISA-driven biomass carbon emissions estimated using slope-corrected AGB were comparable to, although slightly lower than, those derived using AGB obtained from the interpolation method in our manuscript. This agreement confirms the validity of the interpolation method for approximating the carbon density of ISA-contaminated pixels. Given the uncertainties in the results obtained with the slope correction method (detailed below), we have decided to retain the results derived by using the interpolation method in our manuscript, and not to include the slope-corrected AGB results in either the manuscript or the Supplementary Material.

1. Additional explanation of the interpolation method

The interpolation method employed in our study for estimating carbon densities in ISA-contaminated pixels is illustrated in Figure R1a. Suppose that, in 2010, for example, the periphery of a core ISA area (an urban core) expanded to the surrounding 300-m pixels with (according to the ESA CCI land cover product) a land cover type of forest (red-bordered pixels with numbers in circles). However, according to the high-resolution (30 m) ISA products, these pixels contain more than 5% ISA area, and are hence identified as ISA-contaminated pixels whose observed biomass carbon densities are considered to be invalid. Therefore, interpolation is needed to derive their carbon densities under the ‘land homogeneity’ assumption (i.e., without any influence of existing sub-pixel ISA). As an example, consider the ISA-contaminated pixel ②. The carbon density without ISA influence for this pixel is interpolated using the nearest 20 pixels with ISA fraction less than 5% (e.g., ISA-free pixels 1 to 20, the blue-bordered pixels, numbered without circles).

As the referee mentions, it would be preferable to estimate the carbon density in pixel ② using the AGB values of pre-established ISA-contaminated pixels. For example, Figure R1b shows the ISA extent in 2005, where the pixels surrounding the pixel ② (indicated by red-dashed borders) have not yet been influenced by the urban periphery. These pixels are identified as pre-established ISA-contaminated pixels, and, as suggested by the referee, their carbon densities could be used to interpolate the carbon density in pixel ② in 2010. Such an approach can be justified by the fact that these pre-established ISA-contaminated pixels have a lower ISA fraction compared to those in the later years (red-bordered pixels shown in Figure R1a), and are geographically closer to pixel ②, indicating that they are more likely to share similar AGB values to pixel ② than are pixels 1 to 20. Consequently, this method would likely have a higher reliability for the estimation of carbon densities in pixel ② than the method illustrated in Figure R1a. However, as the referee has pointed out, in reality, the carbon density observations for these pre-established pixels (i.e., ① to ⑳ in 2005) are not available, meaning that the method is impossible to apply.

Figure R1 | Schematic illustration of methods for estimating carbon densities for ISA-contaminated pixels using an example of ISA expansion over a forest area. a, ISA extent map in 2010 overlying a carbon density map. The carbon density of an ISA-contaminated pixel (e.g., pixel ②) is interpolated using the nearest 20 ISA-free pixels; **b,** ISA extent map in 2005 overlying a carbon density map. The carbon density of an ISA-contaminated pixel could have been interpolated using the nearby pre-established ISA-contaminated pixels. Squares with blue borders (with numbers without circles) represent forest pixels that are either free from ISA or have an ISA fraction less than 5%; squares with solid red borders (with numbers within circles) indicate ISA-contaminated forest pixels; squares with red-dashed borders represent pre-established ISA-contaminated forest pixels. All pixels have a 300-m spatial resolution with ISA fraction being derived using 30-m resolution ISA products.

2. The alternative approach to correct for ISA influence on observed AGB as proposed by the referee

2.1 Evidence of the influence of ISA on observed AGB

Our analysis primarily relies on the landscape homogeneity assumption, which posits that the biomass carbon density of a vegetated 300m pixel (woody or non-woody) undergoing ISA expansion should be representative of a pixel with a 100% vegetation coverage, excluding any influence of potential sub-pixel ISA. However, in reality, the observed AGB values, for ISA-contaminated forest pixels for example, are a combination of signals from both ISA (assuming zero biomass) and forest, making it necessary to correct for the influence of ISA and to derive the AGB by assuming 100% forest coverage.

A clear hypothesis that emerges from the rationale above is that observed carbon densities in ISA-contaminated pixels are influenced by the presence of sub-pixel ISA, rendering them unreliable for direct application in estimating ISA-driven carbon emissions. We examined the validity of this hypothesis by exploring the relationships between observed AGB and ISA fraction for all 300-m pixels across the globe for different precipitation regimes. This is to avoid any potentially false relationships caused by changes in precipitation, which has a strong influence on biomass carbon density. We used the 30-year annual mean precipitation derived from the WorldClim version 2.1 climate data for 1970-2000. This dataset has a spatial resolution of 1 km, which was resampled to 300 m to match the spatial resolution of the AGB and ISA fraction maps. The annual mean precipitation (P) was divided into 5 categories: All (i.e., the whole globe), $0 < P \leq 400$, $400 < P \leq 800$, $800 < P \leq 1200$, and $P > 1200$.

The ISA fraction of a given 300-m pixel for a given year was calculated as the maximum of the ISA fractions derived from three 30m ISA products: GAUD, GAIA, and GISA. For the Spawn and CCI maps of AGB, the ISA fraction was determined for 2010, while for the Harris AGB map, it was determined for 2000. To ensure consistency in land cover type between the AGB maps and the ESA CCI land cover map, the woody vegetation pixels of the Spawn and CCI AGB maps were identified

by overlaying the map of forest and shrubland derived from the ESA CCI land cover map in 2010. For the Harris AGB map, we used the same method but overlaid the ESA CCI forest and shrubland map for 2000. The non-woody pixels of the Spawn AGB map were identified by overlaying the non-woody land cover types given by the ESA CCI land cover map in 2010.

All 300-m pixels with an ISA fraction larger than 0 were categorized into bins, with increments of 0.01, based on their respective ISA fractions. We then determined the observed woody AGB values from the Spawn, CCI, and Harris maps, as well as the non-woody AGB from the Spawn map, for each ISA-fraction bin. Finally, the mean, median, 25% and 75% percentiles of these values for the different precipitation regimes were calculated. The relationships between the observed AGB and ISA fraction are shown in Figure R2.

The results show that the observed AGB, of both woody and non-woody land, decreases with ISA fraction across different precipitation regimes. The decreasing relationships with increasing fraction are sometimes nonlinear (e.g., for CCI AGB map), similar to the relationships between vegetation index and ISA fraction reported by Zhang et al. (2022) and Zhao et al. (2016). The obvious reduction of AGB with ISA expansion implies that, due to the presence of ISA, the observed AGB cannot be directly applied to estimate ISA-driven biomass emissions, hence validating our hypothesis. Moreover, in general, the mean values of AGB approach zero as the ISA fraction approaches 1, supporting our assumption that full ISA expansion for a given pixel can result in a complete loss of biomass.

Figure R2 | Relationships between observed AGB and ISA fraction under different precipitation regimes. The columns correspond to the 5 annual mean precipitation (P) regimes. The four rows refer, respectively, to the woody AGB maps of Spawn, CCI and Harris, and the non-woody AGB map of Spawn. The dots (triangles) indicate the mean (median) values of observed AGB of 300-m pixels within each 0.01 ISA fraction bin. Green and black indicate woody AGB and non-woody AGB, respectively. The shaded areas illustrate the range of the AGB values (from 25% to 75% percentile).

2.2 Spatial interpolation effectively removes ISA's influence on AGB

To verify that spatial interpolation can effectively remove ISA's influence on AGB, we also investigated the relationships between the interpolated AGB, as applied in the manuscript, and the ISA fraction. Figure R3 presents the results, which are derived following the same method as used to produce Figure R2. The interpolated woody AGB was derived from the average of the three woody AGB maps (i.e., the Spawn, CCI, and Harris map), and the interpolated non-woody AGB was obtained based on the interpolated Spawn map for non-woody vegetation. The 2010 ISA fraction map was applied to both interpolated AGB maps: woody and non-woody.

Figure R3 shows that the reduction of the original AGB due to the presence of ISA is greatly relieved. However, the interpolated woody AGB for pixels with ISA fraction larger than 0.6 (particularly for those larger than 0.8), still exhibits slightly decreasing trends with increasing ISA fraction for the annual mean precipitation category of $400 < P \leq 800$. These woody pixels with ISA fraction larger than 0.6 or 0.8 represent only small proportions of the total number of pixels with ISA fraction greater than 0 (4.8% and 1.8%, respectively, Figure R4; these proportions would be even smaller if the pixels without any ISA were further included). We therefore argue that the potential bias in estimating biomass emissions by using the interpolated AGB would be minimal.

Figure R3 | Relationships between interpolated AGB and ISA fraction under different precipitation regimes. The columns correspond to 5 categories of annual mean precipitation (P). The rows refer, respectively, to the interpolated woody AGB, derived as the average of the three woody AGB maps (i.e., Spawn, CCI and Harris),

and the interpolated non-woody AGB derived from the Spawn map. The dots (triangles) indicate the mean (median) values of the corresponding AGB of 300-m pixels within each 0.01 ISA fraction bin. Green and black indicate woody AGB and non-woody AGB, respectively. The shaded areas illustrate the range of AGB values (from 25% percentile to 75% percentile). Black indicates non-woody AGB while green indicates woody AGB.

Figure R4 | Proportion of pixels in different ISA fraction bins to the total number of pixels with ISA fraction greater than 0 for interpolated woody and non-woody AGB under different precipitation regimes.

2.3 Correct the influence of ISA on AGB by using the slope of ‘AGB~ISA fraction’, following the referee’s suggestion

We also attempted to reconstruct the AGB values with no ISA influence based on the referee’s suggested assumption that biomass carbon loss is proportional to ISA fraction within a 300-m pixel. We first tried to correct the AGB value by dividing the observed AGB by the non-ISA fraction. Unfortunately, the corrected AGB values were found to increase with their original ISA fraction, resulting in extremely large, unrealistic values of estimated AGB when the original ISA fraction exceeds 0.8. Figure R5 shows an example of the issue, in this case when using the Spawn AGB map in a $5^{\circ}\times 5^{\circ}$ tile. This phenomenon clearly demonstrates an unrealistic, overcompensation.

Figure R5 | Relationships between corrected AGB, obtained by dividing the observed AGB by the non-ISA fraction, and the original ISA fraction in a $5^{\circ}\times 5^{\circ}$ tile. a, relationship between corrected AGB and ISA fraction. b, similar to panel a but with a zoomed-in y axis.

Despite these poor results, the referee's suggestion encouraged us to investigate a similar approach, which would possibly have a better outcome: a linear relationship that could possibly be established to derive the slope between observed AGB and ISA fraction, which could then be used to correct the observed AGB and to reconstruct the AGB without ISA influence. We tried to build a linear regression between observed AGB and the ISA fraction within each $5^{\circ} \times 5^{\circ}$ tile, a spatial scale which minimizes the chance of a false relationship between observed AGB and ISA fraction caused by spatial differences. As an example, consider the woody AGB from Spawn in a $5^{\circ} \times 5^{\circ}$ tile located at $0-5^{\circ}\text{E}$, $5-10^{\circ}\text{N}$. We built linear regressions between the originally observed AGB and the ISA fractions by either including or excluding the samples with ISA fraction = 0 (Figure R6). This is done because there are a lot samples with ISA fraction = 0 which would highly influence the regression relationship. However, we found that the ISA fractions could rarely explain the change of the observed AGB and had an almost zero R^2 . In addition, the corrected AGB using the derived slope from the regression including the samples with ISA fraction = 0 shows unrealistic increasing trend with original ISA fraction (Figure R6b, red dots). This means that the method has a low reliability when using the slope derived by fitting linear relationships between AGB and ISA fraction using original data of individual 300-m pixels. However, after grouping the data into different bins ISA of fraction, the R^2 values of the linear regressions have greatly increased. The corrected AGB (detailed below) shows no longer obvious decline with increasing original ISA fraction, indicating a successful removal of the influence of ISA on observed AGB.

Figure R6 | A case of observed and corrected AGB based on the slope correction in a $5^{\circ} \times 5^{\circ}$ tile located at $0-5^{\circ}\text{E}$, $5-10^{\circ}\text{N}$. **a**, the originally observed AGB and the fitting lines using the original or binned linear regressions, with or without the samples with ISA fraction = 0. **b**, the mean corrected AGB derived using the slopes from different fitting methods for each ISA fraction bin. Note that in panel **a**, the three fitted lines (blue, olive-green, and green) almost exactly overlie one another.

Based on the results above, we then decide to use a binned regression approach to smooth the noise associated with observed AGB for individual 300-m pixels. First,

within each $5^{\circ} \times 5^{\circ}$ tile across the globe, all 300-m pixels with an ISA fraction larger than zero were categorized into bins with a constant interval of 0.01. Then, the mean values of observed AGB and ISA fraction for each bin were calculated. Second, simple linear regressions were constructed between the mean AGB and the mean ISA fraction for the four AGB products (i.e., woody AGB of Spawn, CCI, and Harris, and non-woody AGB of Spawn) as:

$$AGB_{obs} = \text{slope} \times ISAF + \text{intercept}$$

where AGB_{obs} is the mean observed AGB, and ISAF is the mean ISA fraction for each ISA fraction bin. For example, in the case shown in Figure R6, the R^2 of the binned regression, irrespective of whether ISA fraction = 0 is included, has been enhanced, from the near zero value of the original regression, to a value larger than 0.5. Further, for the correction of AGB, the linear regressions whose performance met the following two criteria were selected: (1) $p < 0.05$, and (2) a negative slope. The criterion of having negative slopes in the linear regressions is based on the assumption of inevitable biomass loss due to ISA expansion. In the binned regressions, the exclusion or inclusion of the pixels with ISA fraction = 0 did not result in any noticeable changes in the derived slope, intercept, R^2 , or the significance of the linear model (Figure R6). Therefore, we proceeded with the binned regression that includes the pixels with ISA fraction = 0.

Third, for a given $5^{\circ} \times 5^{\circ}$ tile with a valid linear regression model (i.e., meeting the two criteria mentioned above), the corrected AGB values are obtained by subtracting the product of the slope and the ISA fraction from the originally observed AGB of the 300-m pixels. For those tiles without a valid regression model, the originally observed AGB values are retained. Fourth, to make sure that all the 300-m pixels with ISA expansion had valid AGB values, we further interpolated for those pixels with ISA expansion occurring but without valid AGB observations using the 20 nearest valid (corrected from tiles with a valid regression model or original from those without) AGB values with the same land cover types using the inverse distance weighting algorithm, similar to the interpolation method for carbon densities presented in the manuscript. Hereafter, we refer to the AGB processed using the four procedures as the slope-corrected AGB.

Following the method described above, 4 slope-corrected AGB maps were generated (i.e., slope-corrected woody biomass from the Spawn, CCI, and Harris maps, and slope-corrected non-woody biomass from the Spawn map). Then, following the same approach as used to create Figure R2 and Figure R3, the mean, median, and 25% and 75% percentiles of the slope-corrected AGB for each original ISA fraction bin were calculated for different precipitation regimes (Figure R7).

Compared with the originally observed AGB, the slope-corrected AGB also largely mitigated the influence of ISA. Similarly to the interpolated AGB, the corrected AGB shows an obvious decline with increasing ISA fraction larger than 0.6 in the $400 < P \leq 800$ category, but, in general, lower corrected AGB values were obtained

using the slope correction than when using interpolation, with the exception of the $0 < P \leq 400$ category (Figure R8).

Figure R7 | Relationships between the slope-corrected AGB and the fraction of ISA under different precipitation regimes. Each column corresponds to one of the 5 categories of annual mean precipitation (P). The rows refer, respectively, to the woody AGB maps of Spawn, CCI and Harris, and the non-woody AGB map of Spawn. The dots (triangles) indicate the mean (median) values of slope-corrected AGB of 300-m pixels within each 0.01 ISA fraction bin. Green and black denote woody AGB and non-woody AGB, respectively. The shaded areas illustrate the range of AGB values (from 25% percentile to 75% percentile).

Figure R8 | Comparisons between the slope-corrected AGB and interpolated AGB under different precipitation regimes. The columns correspond to the 5 categories of annual mean precipitation (P). The top row relates to the averaged

woody AGB obtained from the Spawn, CCI, and Harris maps, and the bottom row to the non-woody AGB map of Spawn. The dots and triangles indicate the mean values of interpolated AGB and slope-corrected AGB of all 300-m pixels within each 0.01 ISA fraction bin, respectively.

Eventually, we estimated the biomass carbon emissions due to ISA expansion using the three ISA products (i.e., GAUD, GAID, and GISA), the average woody AGB map derived from three slope-corrected woody AGB products (i.e., Spawn, CCI, and Harris), and the slope-corrected non-woody AGB. The quantification of biomass carbon emissions follows the same method as described in the manuscript, with the only difference being the biomass carbon density datasets.

The estimated biomass carbon emissions using slope-corrected AGB were compared with those reported in the manuscript using the interpolated AGB (Figure R9). The biomass emissions estimated using the slope-corrected AGB show similar temporal variations but were 12%~17% lower than those estimated using interpolated AGB. The average biomass emissions from the three ISA products for 1993-2018 were 25.7 Tg C yr⁻¹, slightly lower than the lower limit of the uncertainty (interquartile) range of the biomass emissions using interpolated AGB (31.0_{28.9}^{31.6} Tg C yr⁻¹) for the same period. The lower estimation of biomass carbon emissions is related to the lower biomass densities obtained from the slope-corrected AGB as compared to interpolated AGB. As shown in Figure R6, the bin with an ISA fraction of 0 contains a large number of AGB observations, and its mean AGB is higher than those in other bins. The AGB derived from the interpolation approach was amended using these observed AGB values in pixels without any ISA, probably leading to larger interpolated AGB values than the slope-corrected ones. On the other hand, because the binned regression utilizes the mean values of observed AGB for each bin, the observed AGB in pixels with ISA fraction = 0 are aggregated into a single data point. As a result, their influence on the linear regression model is considerably diminished (see the case in Figure R8). The latter factor may result in an underestimated slope which further results in lower corrected AGB values than those derived using the interpolation method.

Figure R9 | ISA-driven biomass emissions based on the interpolated and slope-corrected AGB. The solid and dashed lines illustrate biomass emissions using interpolated and slope-corrected AGB, respectively.

3. Conclusion

Based on the evidence presented above, we argue that the ISA-driven biomass emissions estimated using interpolated AGB, as in the manuscript, are more reliable than those estimated using slope-corrected AGB. The binned linear regression model lowered the weight of pixels with an ISA fraction of 0, resulting in lower values of slope-corrected AGB compared to those of interpolated AGB. In addition, further uncertainty may be introduced by the binned approach in the linear regression, although the performance of the regression model (i.e., R^2 and p values) was greatly improved compared to the relationship between original AGB and ISA fraction. Nonetheless, the validity of the interpolated AGB method is demonstrated by the evidence that both slope-corrected AGB and interpolated AGB mitigated the influence of ISA on AGB, and the biomass carbon emissions obtained from slope-corrected AGB were comparable and close to the lower limit of the emissions obtained from interpolated AGB. Therefore, we have retained the results calculated by using interpolated AGB in the manuscript. The results obtained by using slope-corrected AGB, as presented in this response letter, have not been included in either the manuscript or the Supplementary Material.

[Comment 2] The primary objective of this study is to quantify the CO₂ release from the original organic matter in plant biomass, surface litter, and mineral soil of previous plant ecosystems that are replaced by ISA, a factor previously overlooked in global budget assessments. The authors' use of the term 'carbon loss' in the title and abstract could potentially lead to confusion about whether or not such carbon loss

includes NPP or NEP lost due to the replacement of ISA. I would recommend that the authors revise the title and abstract to avoid such potential misunderstandings.

[Response] We thank the referee for pointing out this potential confusion. We have changed the title to “Substantial terrestrial carbon emissions from global expansion of impervious surface area”, and changed the term “carbon losses” to “carbon emissions” in both title and abstract. Adding “terrestrial” in the title enhances clarity, as our study focuses on the historical land conversions from terrestrial ecosystems to ISA and its associated carbon emissions, thereby avoiding any potential confusions about carbon losses from the urban environment and those resulting from energy usage in urban areas.

REVIEWERS' COMMENTS

Reviewer #2 (Remarks to the Author):

All my concerns have been addressed. The revised manuscript is ready for publication.